# TAILORING MIXUP TO DATA FOR CALIBRATION

**Quentin Bouniot,**[1,2,3,4] **Pavlo Mozharovskyi**[1] **& Florence d'Alché-Buc**[1]
[1]LTCI, Télécom Paris, Institut Polytechnique de Paris, France,
[2]Technical University of Munich,   [3]Helmholtz Munich,
[4]Munich Center for Machine Learning (MCML)

## ABSTRACT

Among all data augmentation techniques proposed so far, linear interpolation of training samples, also called *Mixup*, has found to be effective for a large panel of applications. Along with improved predictive performance, Mixup is also a good technique for improving calibration. However, mixing data carelessly can lead to manifold mismatch, i.e., synthetic data lying outside original class manifolds, which can deteriorate calibration. In this work, we show that the likelihood of assigning a wrong label with mixup increases with the distance between data to mix. To this end, we propose to dynamically change the underlying distributions of interpolation coefficients depending on the similarity between samples to mix, and define a flexible framework to do so without losing in diversity. We provide extensive experiments for classification and regression tasks, showing that our proposed method improves predictive performance and calibration of models, while being much more efficient.

## 1 INTRODUCTION

The *Vicinal Risk Minimization (VRM)* principle (Chapelle et al., 2000) improves on the well-known *Empirical Risk Minimization (ERM)* (Vapnik, 1998) for training deep neural networks by drawing virtual samples from a vicinity around true training data. This data augmentation principle is known to improve the generalization ability of deep neural networks when the number of observed data is small compared to the task complexity. In practice, the method of choice to implement it relies on hand-crafted procedures to mimic natural perturbations (Yaeger et al., 1996; Ha & Bunke, 1997; Simard et al., 2002). However, one counterintuitive but effective and less application-specific approach for generating synthetic data is through interpolation, or mixing, of two or more training data.

The process of interpolating between data have been discussed multiple times before (Chawla et al., 2002; Wang et al., 2017; Inoue, 2018; Tokozume et al., 2018), but *Mixup* (Zhang et al., 2018) represents the most popular implementation and continues to be studied in recent works (Pinto et al., 2022; Liu et al., 2022b; Wang et al., 2023). Ever since its introduction, it has been a widely studied data augmentation technique spanning applications to *image classification and generation* (Zhang et al., 2018), *semantic segmentation* (Franchi et al., 2021; Islam et al., 2023), *natural language processing* (Verma et al., 2019), *speech processing* (Meng et al., 2021), *time series and tabular regression* (Yao et al., 2022a) or *geometric deep learning* (Kan et al., 2023), to that extent of being now an integral component of competitive state-of-the-art training settings (Wightman et al., 2021). The idea behind *Mixup* can be seen as an efficient approximation of *VRM*, by using a linear interpolation of data points from within the same batch to reducing computation (Zhang et al., 2018).

The process of Mixup as a data augmentation during training can be roughly separated in three phases: (i) selecting tuples (most often pairs) of points to mix together, (ii) sampling coefficients that will govern the interpolation to generate synthetic points, (iii) applying a specific interpolation procedure between the points weighted by the coefficients sampled. Methods in the literature have mainly focused on the first and third phases, *i.e.*, the process of sampling points to mix through predefined criteria (Hwang et al., 2022; Yao et al., 2022a;b; Palakkadavath et al., 2022; Teney et al., 2023) and on the interpolation itself, by applying sophisticated and application-specific functions (Yun et al., 2019; Franchi et al., 2021; Venkataramanan et al., 2022; Kan et al., 2023). On the other hand, these *interpolation coefficients*, when they exist, are always sampled from the same distribution throughout training. Recent works have shown that mixing carelessly different points can result in *manifold*

*intrusions*, i.e. mixed examples colliding with other real data manifolds (Guo et al., 2019; Baena et al., 2022; Chidambaram et al., 2021), or *label noise*, i.e. mixed labels differing from ground-truth assignments, which can hurt generalization (Yao et al., 2022a; Liu et al., 2023), while mixing similar points helps in diversity (Chawla et al., 2002; Dablain et al., 2022). Furthermore, several previous work have highlighted a trade-off between good predictive performance and calibration in Mixup (Thulasidasan et al., 2019; Pinto et al., 2022; Wang et al., 2023).

In this work, we show that the distance between the data used in the mixing impacts the likelihood of assigning a noisy label, which occurs from *manifold mismatch*, i.e. mixed samples lying outside the class manifolds of the original data used in the mixing operation. Manifold mismatch can then lead to manifold intrusion, if the mixed sample *lies* in a different class manifold, or label noise if the mixed sample is *closer* to a different class manifold. Then, we propose an efficient and flexible framework to take the similarity between the points into account by influencing the interpolation coefficients. A high similarity should result in a strong interpolation, while a low similarity should lead to small changes. Consequently, controlling the interpolation through the distance of the points to mix improves calibration by reducing manifold mismatch. Our contributions [1] in this paper are:

- We show that the likelihood of assigning a noisy label is increasing with the distance between points, and taking into account distance when mixing can improve calibration. Mixing only similar data leads to better calibration than including dissimilar ones.

- We present a flexible framework to dynamically change the distributions of interpolation coefficients. We apply a **similarity kernel** that takes into account the distance between points to select a parameter for the distribution tailored to each pair to mix. The underlying distribution of interpolation is warped to be stronger for similar points and weaker otherwise.

- We quantitatively ascertain the effectiveness of our **Similarity Kernel Mixup** with extensive experiments on multiple datasets, from image classification to regression tasks, and multiple deep neural network architectures. Our approach achieves better *calibration* while improving accuracy. Additionally, we highlight the **efficiency** of our method, obtaining competitive results with less computation per iteration, and reaching the best performance globally faster.

## 2 RELATED WORK

### 2.1 DATA AUGMENTATION BASED ON MIXING DATA

**Non-linear interpolation**   Non-linear combinations are mainly studied for dealing with image data. Instead of a naive linear interpolation between two images, the augmentation process is done using more complex non-linear functions, such as cropping, patching and pasting images together (Takahashi et al., 2019; Summers & Dinneen, 2019; Yun et al., 2019; Kim et al., 2020; Hendrycks et al., 2020) or through subnetworks (Ramé et al., 2021; Liu et al., 2022b; Venkataramanan et al., 2022). Not only are these non-linear operations focused on images, but they generally introduce a significant computational overhead compared to the simpler linear one (Zhu et al., 2020; Li et al., 2022). The recent *R-Mixup* (Kan et al., 2023), on the other hand, considers other Riemannian geodesics rather than the Euclidean straight line for graphs, but is also computationally expensive. Furthermore, Oh & Yun (2023) has recently shown that learning with linear mixup leads to more meaningful decision boundaries.

**Linear interpolation**   Mixing samples online through linear interpolation represents the most efficient and general technique compared to the ones presented above (Zhang et al., 2018; Inoue, 2018; Tokozume et al., 2018). Among these different approaches, combining data from the same batch also avoids additional samplings. Several follow-up works extend Mixup from different perspectives. *Manifold Mixup* (Verma et al., 2019) interpolates data in the feature space, *Remix* (Chou et al., 2020) separates the interpolation in the label space and the input space. Notably, *AdaMixUp* (Guo et al., 2019) learns to predict a mixing policy to apply to avoid manifold intrusion using two additional parallel models with intrusion losses. This leads to a more complex training and includes more parameters to learn. Furthermore, the predicted range is very narrow around 0.5, reducing diversity of the mixed samples, and the training includes non-mixed samples, making the approach similar to the recent *Regmixup* (Pinto et al., 2022), described below. *Local Mixup* (Baena et al., 2022) is

---

[1]Code is available at https://github.com/qbouniot/sim_kernel_mixup

another approach interested on the manifold intrusion problem, by weighting the loss function with the distance between the points using K-nearest neighbors graphs. However, this results in completely discarding the interpolation with points that are far away, losing potential augmentation directions and, therefore, in diversity. Moreover, these two methods are focusing on improving generalization, while we are interested in the more general problem of manifold mismatch to improve *calibration*.

**Selecting points** A recent family of methods applies an online linear combination on specifically selected pairs of examples (Yao et al., 2022a;b; Hwang et al., 2022; Palakkadavath et al., 2022; Teney et al., 2023), across classes (Yao et al., 2022b) or across domains (Yao et al., 2022b; Palakkadavath et al., 2022; Tian et al., 2023). These methods achieve impressive results on distribution shift and out-of-distribution generalization (Yao et al., 2022b), but recent theoretical developments have shown that much of the improvements are linked to a resampling effect from the restrictions in the selection process, and are unrelated to the mixing operation (Teney et al., 2023). These selective criteria also induce high computational overhead. One related approach is *C-Mixup* (Yao et al., 2022a), that fits a Gaussian kernel on the labels distance between points in regression tasks. Then points to mix together are sampled from the full training set according to the learned Gaussian density. However, the Gaussian kernel is computed on all the data before training, which is difficult when there is a lot of data and no explicit distance between them. *k-Mixup* (Greenewald et al., 2023) selects the pairs to mix based on Optimal Transport distance between two different batch of data.

## 2.2 CALIBRATION IN CLASSIFICATION AND REGRESSION

*Calibration* is a metric to quantify uncertainty, measuring the difference between a model's confidence in its predictions and the actual probability of those predictions being correct. We refer the reader to Appendix C for a presentation of common calibration metrics used.

**In classification** Modern deep neural network for image classification are now known to be *overconfident* leading to *miscalibration* (Guo et al., 2017). One can rely on *temperature scaling* (Guo et al., 2017) to improve calibration *post-hoc*, or using different techniques during learning such as *ensembling* (Lakshminarayanan et al., 2017; Wen et al., 2021; Laurent et al., 2022), *explicit penalties* (Pereyra et al., 2017; Kumar et al., 2018; Moon et al., 2020; Cheng & Vasconcelos, 2022), or *implicit* ones (Müller et al., 2019; Lin et al., 2017; Mukhoti et al., 2020; Liu et al., 2022a), including notably through *mixup* (Thulasidasan et al., 2019; Pinto et al., 2022; Noh et al., 2023). One should note that the ordering of calibration results can change after temperature scaling (Ashukha et al., 2020).

**Calibration-driven Mixup methods** The problem of the trade-off between predictive performance and calibration with Mixup has been extensively studied in previous works (Thulasidasan et al., 2019; Zhang et al., 2022; Pinto et al., 2022; Wang et al., 2023). Notably, Wang et al. (2023) observed that calibration using Mixup can be degraded after temperature scaling. Therefore, they proposed another improvement of mixup, *MIT* (Wang et al., 2023), by generating two sets of mixed samples and then deriving their correct label. *AugMix* (Hendrycks et al., 2020), *NFM* (Lim et al., 2022) and *NoisyMix* (Erichson et al., 2024) add data-augmentation and noise before the mixing operation to improve robustness. *RegMixup* (Pinto et al., 2022) considers Mixup as a regularization term, using stronger interpolation on every pair. Finally, *RankMixup* (Noh et al., 2023) uses the interpolation coefficients from multiple mixed pairs as an additional supervisory signal for ranking of confidence. We propose a more efficient method to achieve a good trade-off between accuracy and calibration with Mixup.

**In regression** The problem of calibration in deep learning has also been studied for regression tasks (Kuleshov et al., 2018; Song et al., 2019; Laves et al., 2020; Levi et al., 2022), where it is more complex as we lack a simple measure of predictive confidence. In this case, regression models are usually evaluated under the variational inference framework with Monte Carlo (MC) Dropout (Gal & Ghahramani, 2016) to quantify confidence.

In our work, we use a *similarity kernel* to mix more strongly similar data and avoid mixing less similar ones, leading to preserving label quality and confidence of the network. As opposed to all other methods discussed above, we also show the flexibility of our approach by its effectiveness on both classification and regression tasks. We detail our framework, the similarity used and the intuition behind below.

## 3 SIMILARITY KERNEL MIXUP

First, we define the notations and elaborate on the learning conditions that will be considered throughout the paper. Let $\mathcal{D} = \{(\mathbf{x}_i, y_i)\}_{i=1}^N = (\mathbf{X}, \mathbf{y}) \in \mathbb{X}^N \times \mathbb{Y}^N \subset \mathbb{R}^{d_1 \times N} \times \mathbb{R}^N$ be the training

dataset. We want to learn a *model* $f$ parameterized by $\theta \in \Theta \subset \mathbb{R}^p$, that predicts $\hat{y} := f(\mathbf{x})$ for any $\mathbf{x} \in \mathbb{X}$. For classification tasks, we have $\mathbb{Y} = \{1, \ldots, M\}$, where $M$ is the number of classes, and we further assume that the model $f$ can be separated into an encoder part $h : \mathbb{X} \to \mathbb{R}^{d_2}$ and classification weights $\mathbf{w} \in \mathbb{R}^{d_2 \times M}$, with $d_2$ the embedding dimension, such that $\forall \mathbf{x} \in \mathbb{X}, f(x) = \mathbf{w}^T h(x)$ and $\hat{y} := \arg\max_{m \in \{1, \ldots, M\}} \text{softmax}(f(x))_m$. To learn our model, we optimize the *weights* of the model $\theta$ in a stochastic manner, by repeating the minimization process of the empirical risk computed on *batch* of data $\mathcal{B}_t = \{(\mathbf{x}_i, y_i)\}_{i=1}^n$ sampled from the training set, for $t \in \{1, \ldots, T\}$ iterations.

With *Mixup* (Zhang et al., 2018), at each iteration $t$, the empirical risk is computed on an augmented batch of data $\tilde{\mathcal{B}}_t = \{(\tilde{\mathbf{x}}_i, \tilde{y}_i)\}_{i=1}^n$, such that $\tilde{\mathbf{x}}_i := \lambda_t \mathbf{x}_i + (1 - \lambda_t) \mathbf{x}_{\sigma_t(i)}$ and $\tilde{y}_i := \lambda_t y_i + (1 - \lambda_t) y_{\sigma_t(i)}$, with $\lambda_t \sim \texttt{Beta}(\alpha, \alpha)$ and $\sigma_t \in \mathfrak{S}_n$ a random permutation of $n$ elements sampled uniformly. Thus, each input is mixed with another input randomly selected from the same batch, and $\lambda_t$ represents the strength of the interpolation between them. Besides simplicity, mixing elements within the batch significantly reduces both memory and computation costs.

In the following parts, we introduce a more general extension of this framework using warping functions, that spans different variants of *Mixup*, while preserving its efficiency. First, we show and illustrate how *Mixup* can lead to *manifold mismatch* and its impact on confidence scores.

## 3.1 Manifold Mismatch

We consider the general setup of *manifold learning* (Vural & Guillemot, 2018). We assume $\mathbb{X} \subset \mathcal{H}$ with $\mathcal{H}$ being a Hilbert space and consider a classification problem with $M > 2$ classes, where samples of each class $m \in \{1, 2, \ldots, M\}$ are drawn from a probability measure $\nu_m$ such that $\nu_m$ has bounded support $\mathcal{M}_m \subset \mathbb{X}$. We further assume that the classes are separated, such that all samples belong to a single class and manifolds are disjoints. Then, $\forall (\mathbf{x}_i, y_i) \in \mathcal{D}, y_i \in \{1, \ldots, M\}$ and $\mathbf{x}_i \sim \nu_{y_i}$. In this setting, we show the following results, with proof in Appendix F:

**Theorem 3.1.** *For any pair of manifold $\mathcal{M}_i, \mathcal{M}_j$, there exists $\mathbf{x}_k, \mathbf{x}_l \in \mathcal{M}_i \cup \mathcal{M}_j$, and $\lambda_1, \lambda_2 \in [0, 1]$, $\lambda_1 > \lambda_2$, such that :*

*(i) $\forall \lambda \in ]\lambda_2, \lambda_1[$, $\tilde{\mathbf{x}}(\lambda) = \lambda \mathbf{x}_k + (1 - \lambda) \mathbf{x}_l$ does not belong to the same manifold as $\mathbf{x}_k$ and $\mathbf{x}_l$.*

*(ii) $\|\tilde{\mathbf{x}}(\lambda_1) - \tilde{\mathbf{x}}(\lambda_2)\|_{\mathcal{H}} = |\lambda_1 - \lambda_2| \|\mathbf{x}_k - \mathbf{x}_l\|_{\mathcal{H}}$.*

In other words, for any pair of class, we can find pairs of points where their convex combination can fall in a region of the space (in the line segment) that belongs to neither of the original manifolds. These regions are delimited by the borders of the manifolds ($\tilde{\mathbf{x}}(\lambda_1)$ and $\tilde{\mathbf{x}}(\lambda_2)$) on the line segment, and can be seen as regions of uncertainty with respect to the true underlying manifold. In practice, convex combinations of points that falls in such regions leads to *manifold mismatch*, as they could either belong to a different class manifold or to no class manifold at all.

Now that we have shown the existence of combinations leading to *manifold mismatch*, we introduce a corollary of the theorem presented in Liu et al. (2023). We consider $X, X'$ random variables associated to the inputs, and $\tilde{X} = (1 - \lambda)X + \lambda X'$, for a fixed $\lambda \in [0, 1]$. The ground truth conditional distribution of the labels $Y$ is expressed as a vector-valued function $g : \mathbb{X} \to \mathbb{R}^M$, such that $P(Y = m | X = x) = g_m(x)$, for each dimension $m \in \{1, \ldots, M\}$. The mixup-induced label can then be assigned based on ground truth $\tilde{Y}^* = \arg\max_m g_m(\tilde{X})$, or from the combination of conditional distribution $\tilde{Y} = \arg\max_m [(1 - \lambda)g_m(X) + \lambda g_m(X')]$. With that, we can say that the mixup label is *noisy* when the two assignments disagree, i.e. when $\tilde{Y}^* \neq \tilde{Y}$.

**Theorem 3.2.** *For any fixed $X, X'$ and $\tilde{X}$ related by $\tilde{X} = (1 - \lambda)X + \lambda X'$ for a fixed $\lambda \in [0, 1]$, and for samples $(\mathbf{x}, y), (\mathbf{x}', y') \in \mathcal{D}$, with $\tilde{\mathbf{x}} = (1 - \lambda)\mathbf{x} + \lambda \mathbf{x}'$, the probability of assigning a noisy label is lower bounded by*

$$P(\tilde{Y}^* \neq \tilde{Y} | \tilde{X} = \tilde{\mathbf{x}}) \geq \frac{\min(\lambda, (1 - \lambda))}{R} W_1(\mathcal{M}_y, \mathcal{M}_{y'}) \tag{1}$$

*where $W_1(\cdot, \cdot)$ is the Wasserstein metric, and $R = \sup_{\mathbf{x}, \mathbf{x}' \in \mathbb{X}} \|\mathbf{x} - \mathbf{x}'\|_{\mathcal{H}}$ is the radius of the space (see Appendix F.2).*

This result tells us that the probability of assigning a noisy label is lower bounded by the distance between the original manifolds. Thus, to avoid noisy labels, we want to reduce the probability that

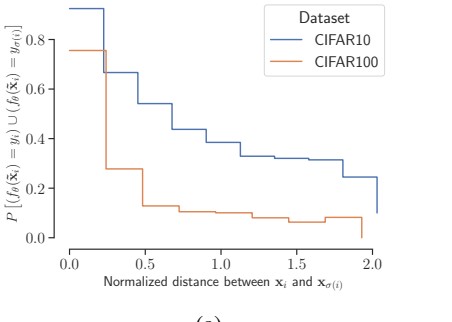 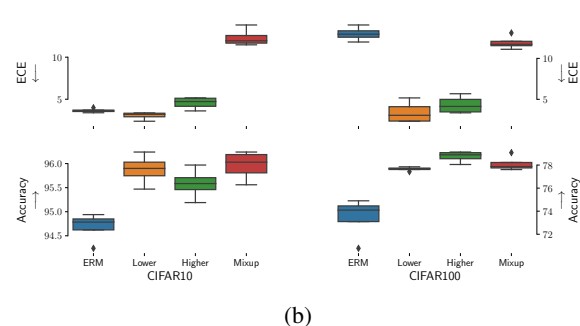

(a)                                (b)

Figure 1: **(a)** Probability that predicted label of mixed samples corresponds to the label of either of the two points used for mixing, depending on the distance between the two points. **(b)** Performance (Accuracy in %, *higher is better*, **bottom**) and calibration (ECE, *lower is better*, **top**) comparison with Resnet34 on CIFAR10 (**left**) and CIFAR100 (**right**) datasets. We compare results when mixing only elements with distance **Higher** (in *green*) than the median, and **Lower** (in *orange*) than the median of all pairwise distances within each batch.

$\lambda = 0.5$ when the original manifolds are far from each other. Furthermore, the distance between two points roughly informs about the distance between the two manifolds.

From the above discussion, we assume that in practice, the higher the distance between two points, the less likely their convex combination will fall near the original manifolds, and thus, the more likely a model would assign a different label than the original labels of the two points. We verify this in the following experiments presented in Figure 1a. Using a Resnet18 (He et al., 2016) trained with ERM respectively on CIFAR10 or CIFAR100 (Krizhevsky et al., 2009), we generate 10000 mixed samples $\tilde{\mathbf{x}}_i$ from data of each test set. Each interpolation coefficient used to obtain $\tilde{\mathbf{x}}_i$ is sampled from a uniform distribution between 0 and 1. We then measure the frequency that the model assigns the same label to $\tilde{\mathbf{x}}_i$ than the label of either of the points in the pair used to mix ($y_i$ or $y_{\sigma(i)}$), with respect to the distance between the points in the pair. As can be seen, the frequency of assigning the same label decreases with the distance. This confirms that the uncertainty on the label of the mixed samples directly depends in practice on the distance between the two points.

Then, we conduct an empirical analysis to compare accuracy and calibration metrics using the classic *ERM*, the original *Mixup*, and *a modified version of Mixup* that selects pairs to mix according to a given quantile of the overall pairwise distances within the batch. To have equivalent proportions of possible data to mix with (*same diversity*), we analyze results when mixing only pairs of points with distances lower than a given quantile $q$, and higher than the quantile $q' = 1 - q$. More specifically, we show in Figure 1b the results for $q = 0.5$ (the median) of the overall pairwise distances within the batch, using a Resnet34 (He et al., 2016) on CIFAR10 and CIFAR100 (Krizhevsky et al., 2009) datasets. Detailed values for this experiment with different quantiles $q$ can be found in Appendix D, and implementation details in Section 4. First, we observe that Mixup improves upon ERM's accuracy, but can degrade calibration depending on the dataset, which is consistent with findings from Wang et al. (2023). Then, when selecting pairs according to distance, a sufficiently high proportion of data to mix is necessary to improve accuracy ($q \geq 0.5$). Finally, mixing data with lower distances achieves a better calibration as opposed to mixing data with higher distances. These results show that there is a trade-off between adding diversity by increasing the proportion of elements to mix, and uncertainty by mixing elements far from each other. Furthermore, it shows that we cannot restrict pairs to mix by selecting data solely based on distance, as it can degrade performance by reducing diversity of synthetic samples. To better control this trade-off with Mixup, we propose to tailor *interpolation coefficients* to the training data, from the distance between the points to mix, such that all points and directions can be considered for mixing.

### 3.2 WARPED MIXUP

To dynamically change the interpolation depending on the similarity between points, we rely on *warping functions* $\omega_\tau$, to *warp* interpolation coefficients $\lambda_t$ at every iteration $t$ depending on the

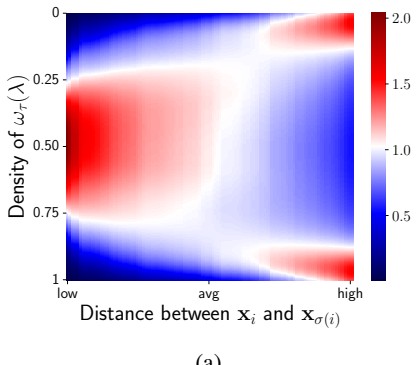
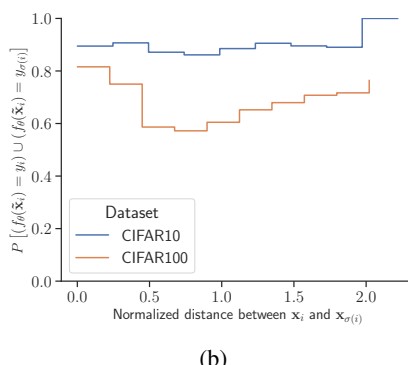

(a)                                                    (b)

Figure 2: **(a)** Density of interpolation coefficients $\omega_\tau(\lambda)$ *after warping with the similarity kernel* depending on the distance between pairs. **(b)** Probability that predicted label for mixed samples with SK mixup corresponds to the label of either of the two points used for mixing.

parameter $\tau$. These functions $\omega_\tau$ are *bijective transformations* from $[0, 1]$ to $[0, 1]$ defined as such:

$$\omega_\tau(\lambda_t) = I_{\lambda_t}^{-1}(\tau, \tau) \quad \text{with} \quad I_{\lambda_t} = \int_0^{\lambda_t} \frac{u^{\tau-1}(1-u)^{\tau-1}}{B(\tau, \tau)} du, \tag{2}$$

where $I_{\lambda_t}$ is the *regularized incomplete beta function*, which is the *cumulative distribution function (CDF) of the Beta distribution*, $B(\tau, \tau)$ is a normalization constant and $\tau \in \mathbb{R}_+^*$ is the *warping parameter* that governs the strength and direction of the warping. Our motivation behind such $\omega_\tau$ is to preserve the same type of distribution after warping, *i.e.*, Beta distributions with symmetry around $0.5$, through *Inverse Transform Sampling* (see Appendix F for the proof):

**Proposition 3.3.** *Let $\lambda \sim \mathcal{U}([0, 1])$. Then $\omega_\tau(\lambda) \sim \texttt{Beta}(\tau, \tau)$ for any $\tau > 0$.*

One should note that according to Proposition 3.3, the initial interpolation parameter $\lambda$ should follow a *uniform distribution* for $\omega_\tau(\lambda)$ to follow a $\texttt{Beta}(\tau, \tau)$ distribution. Thus, in the remaining of the paper, we always draw $\lambda$ according to a uniform distribution (or equivalently, a $\texttt{Beta}(1, 1)$), which has the additional benefit of *removing $\alpha$ as a hyperparameter*. We thus extend the Mixup framework:

$$\lambda_t \sim \texttt{Beta}(1, 1), \qquad \begin{aligned} \tilde{\mathbf{x}}_i &:= \omega_\tau(\lambda_t)\mathbf{x}_i + (1 - \omega_\tau(\lambda_t))\mathbf{x}_{\sigma_t(i)} \\ \tilde{y}_i &:= \omega_\tau(\lambda_t)y_i + (1 - \omega_\tau(\lambda_t))y_{\sigma_t(i)}. \end{aligned} \tag{3}$$

In addition to providing a general framework, using warping functions is more computationally efficient in practice, which we discuss in details in Appendix G. In the following part, we present our method to select the right $\tau$ depending on the data to mix, to dynamically change the distribution of the interpolation coefficients.

## 3.3 SIMILARITY KERNEL

Recall that our main goal is to apply stronger interpolation between similar points, and reduce interpolation otherwise. Since the behavior of Beta distributions and $\omega_\tau$ are logarithmic with respect to $\tau$, therefore, the parameter $\tau$ should be exponentially correlated with the distance, with a symmetric behavior around 1. To this end, we define a class of *similarity kernels*, based on a normalized and centered Gaussian kernel, that outputs the correct warping parameter for the given pair of points. Given a batch of data $\mathbf{x} = \{\mathbf{x}_i\}_{i=1}^n \in \mathbb{R}^{d \times n}$, the index of the first element in the mix $i \in \{1, \ldots, n\}$, along with the permutation $\sigma \in \mathfrak{S}_n$ to obtain the index of the second element, we compute the following *similarity kernel*:

$$\tau(\mathbf{x}, i, \sigma; \tau_{\max}, \tau_{\text{std}}) = \tau_{\max} \exp\left(-\frac{\bar{d}_n(\mathbf{x}_i, \mathbf{x}_{\sigma(i)}) - 1}{2\tau_{\text{std}}^2}\right), \tag{4}$$

$$\text{with} \quad \bar{d}_n(\mathbf{x}_i, \mathbf{x}_{\sigma(i)}) = \frac{\|\mathbf{x}_i - \mathbf{x}_{\sigma(i)}\|_2^2}{\frac{1}{n}\sum_{j=1}^n \|\mathbf{x}_j - \mathbf{x}_{\sigma(j)}\|_2^2}, \tag{5}$$

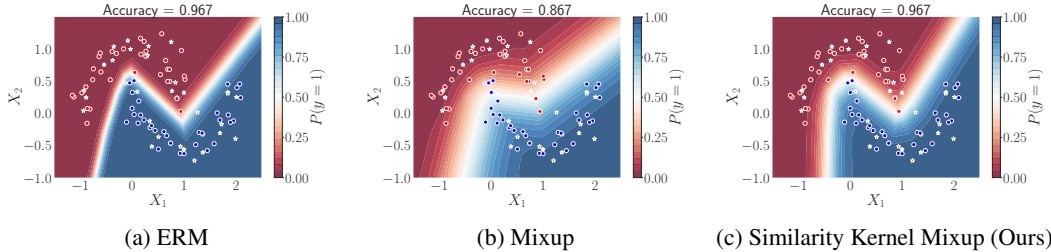

(a) ERM      (b) Mixup      (c) Similarity Kernel Mixup (Ours)

Figure 3: Decision frontiers and data used during training (*circles*) and testing (*stars*) for (a) ERM, (b) Mixup, and (c) our Similarity Kernel Mixup, on Moons toy dataset.

where $\bar{d}_n$ is the squared $L_2$ distance rescaled by the mean distance over the batch, and $\tau_{\max}, \tau_{\mathrm{std}}$ are respectively the *amplitude* and *standard deviation (std)* of the Gaussian, which are hyperparameters of the similarity kernel. The amplitude $\tau_{\max}$ governs the strength of the interpolation in average, and $\tau_{\mathrm{std}}$ the extent of mixing. In practice, we found that these two parameters could be tuned separately, and that choosing a good $\tau_{\mathrm{std}}$ is more important than $\tau_{\max}$ for calibration. This framework allows measuring similarity between points in any space that can represent them. More specifically, for classification, we use the $L_2$ distance between embeddings, *i.e.*, $\bar{d}_n(h(\mathbf{x}_i), h(\mathbf{x}_{\sigma(i)}))$, while for regression, we use the distance between labels, *i.e.*, $\bar{d}_n(y_i, y_{\sigma(i)})$, following Yao et al. (2022a).

Our motivation behind this kernel is to have $\tau > 1$ when the two points to mix are similar, *i.e.*, the distance is lower than average, to increase the mixing effect, and $\tau < 1$ otherwise, to reduce the mixing. Figure 2a illustrates the evolution of the density of *warped* interpolation coefficients $\omega_\tau(\lambda)$, depending on the distance between the points to mix. Close distances *(left part of the heatmap)* induce strong interpolations, while far distances *(right part of the heatmap)* reduce interpolation. Using this similarity kernel to find the correct $\tau$ to parameterize the *Beta distribution* defines our full *Similarity Kernel (SK) Mixup* framework. A detailed algorithm of the training procedure can be found in Appendix E, and an in-depth discussion about warping and similarity measures in Appendix K.

In Figure 2b, we reproduce the experiments presented in Figure 1a, but using our SK Mixup to generate mixed samples, instead of a uniform distribution. We can see that governing interpolation by the distance reduces likelihood of manifold mismatch even when the samples are far away. Then, we also illustrate and compare behaviors of our method on the *Moons* toy problem in Figure 3. We observe that a model with Mixup *(middle)* can lead to worst confidence and performance compared to standard ERM *(left)*, due to manifold mismatch. Since the moons are intertwined and non-convex, linear interpolations from the same moon can lie in the other. Using our SK Mixup approach *(right)*, we recover the accuracy and achieve more meaningful confidence scores.

## 4 EXPERIMENTS

### 4.1 PROTOCOLS

**Image Classification** We follow experimental settings from previous works (Liu et al., 2022a; Pinto et al., 2022; Wang et al., 2023; Noh et al., 2023) and evaluate our approach on CIFAR-10, CIFAR-100 (Krizhevsky et al., 2009), Tiny-Imagenet (Deng et al., 2009) and Imagenet (Russakovsky et al., 2015) datasets for In-Distribution (ID) performance and calibration. We evaluate models trained on CIFAR10 and CIFAR100 on CIFAR10-C and CIFAR100-C (Hendrycks & Dietterich, 2018) for covariate shift robustness, and models trained on ImageNet on Imagenet-R (Hendrycks et al., 2021a) and ImageNet-A (Hendrycks et al., 2021b) for Out-Of-Distribution (OOD) robustness, using Resnet34, Resnet50, Resnet101 (He et al., 2016) and ViT (Dosovitskiy et al., 2021) architectures. We evaluate calibration using ECE and AECE (Naeini et al., 2015; Guo et al., 2017), negative log likelihood (NLL) (Hastie et al., 2009) and Brier score (Brier, 1950), and also after finding the optimal temperature on the validation set through Temperature Scaling (TS) (Guo et al., 2017). Results are reproduced and averaged over 4 different random runs, and we report standard deviation.
**Regression** Here again, we follow settings of previous work on regression (Yao et al., 2022a). We evaluate performance on Airfoil (Kooperberg, 1997), Exchange-Rate and Electricity (Lai et al., 2018) datasets using Mean Averaged Percentage Error (MAPE), along with Uncertainty Calibration

Table 1: Comparison of Performance (Accuracy in %) and calibration (ECE, Brier, NLL) with Resnet34 on CIFAR10 and CIFAR100. Best in **bold**, second best underlined.

| Methods | $\alpha$ | $\tau_{std}$ | CIFAR10 | | | | CIFAR100 | | | |
|---|---|---|---|---|---|---|---|---|---|---|
| | | | Accuracy ($\uparrow$) | ECE ($\downarrow$) | Brier ($\downarrow$) | NLL ($\downarrow$) | Accuracy ($\uparrow$) | ECE ($\downarrow$) | Brier ($\downarrow$) | NLL ($\downarrow$) |
| ERM | – | – | $94.69 \pm 0.27$ | $3.65 \pm 0.22$ | $8.90 \pm 0.40$ | $24.57 \pm 1.04$ | $73.47 \pm 1.59$ | $13.00 \pm 0.17$ | $39.56 \pm 2.19$ | $121.06 \pm 7.92$ |
| Mixup | 1 | – | $95.97 \pm 0.27$ | $12.30 \pm 0.92$ | $8.54 \pm 0.64$ | $25.68 \pm 1.53$ | $78.11 \pm 0.57$ | $11.92 \pm 0.73$ | $32.82 \pm 0.87$ | $95.76 \pm 2.43$ |
| | 0.5 | – | $95.71 \pm 0.26$ | $6.85 \pm 1.42$ | $7.46 \pm 0.54$ | $21.52 \pm 1.70$ | $77.14 \pm 0.67$ | $8.06 \pm 1.19$ | $32.78 \pm 1.11$ | $96.59 \pm 3.78$ |
| | 0.1 | – | $95.37 \pm 0.22$ | $2.02 \pm 0.4$ | $7.48 \pm 0.46$ | $17.99 \pm 1.21$ | $76.01 \pm 0.62$ | $\underline{3.06 \pm 0.74}$ | $33.51 \pm 0.61$ | $94.40 \pm 1.69$ |
| Mixup IO | 1 | – | $95.16 \pm 0.22$ | $\underline{1.92 \pm 0.06}$ | $7.51 \pm 0.33$ | $16.80 \pm 0.56$ | $74.44 \pm 0.49$ | $9.90 \pm 0.65$ | $36.94 \pm 0.40$ | $106.38 \pm 1.47$ |
| | 0.5 | – | $95.31 \pm 0.17$ | $\underline{2.19 \pm 0.18}$ | $7.40 \pm 0.24$ | $17.0 \pm 0.69$ | $74.45 \pm 0.6$ | $10.51 \pm 0.75$ | $37.2 \pm 0.77$ | $108.05 \pm 2.74$ |
| | 0.1 | – | $95.12 \pm 0.21$ | $2.74 \pm 0.21$ | $7.84 \pm 0.32$ | $19.04 \pm 0.95$ | $74.21 \pm 0.46$ | $8.09 \pm 0.20$ | $36.39 \pm 0.53$ | $103.08 \pm 2.02$ |
| SK Mixup *(Ours)* | 1 | 0.2 | $96.29 \pm 0.07$ | $\mathbf{1.81 \pm 0.24}$ | $\underline{6.48 \pm 0.13}$ | $\mathbf{16.14 \pm 0.28}$ | $78.13 \pm 0.52$ | $\mathbf{2.72 \pm 0.73}$ | $31.33 \pm 0.79$ | $\mathbf{85.96 \pm 2.18}$ |
| | | 0.4 | $\mathbf{96.42 \pm 0.24}$ | $3.04 \pm 0.26$ | $\mathbf{6.24 \pm 0.43}$ | $\underline{16.38 \pm 1.01}$ | $\underline{78.86 \pm 0.83}$ | $7.62 \pm 0.82$ | $\underline{30.87 \pm 1.31}$ | $88.32 \pm 3.92$ |
| | | 0.6 | $\underline{96.36 \pm 0.09}$ | $4.89 \pm 0.56$ | $6.55 \pm 0.12$ | $18.17 \pm 0.51$ | $78.63 \pm 0.27$ | $7.98 \pm 0.97$ | $31.21 \pm 0.31$ | $89.57 \pm 0.98$ |
| | | 0.8 | $96.00 \pm 0.41$ | $5.83 \pm 0.31$ | $7.15 \pm 0.55$ | $19.88 \pm 1.23$ | $\mathbf{79.12 \pm 0.52}$ | $8.62 \pm 0.78$ | $\mathbf{30.68 \pm 0.55}$ | $\underline{86.59 \pm 4.0}$ |
| | | 1.0 | $96.25 \pm 0.07$ | $5.9 \pm 0.8$ | $6.89 \pm 0.25$ | $19.38 \pm 0.78$ | $78.51 \pm 0.94$ | $8.73 \pm 1.41$ | $31.45 \pm 1.28$ | $90.42 \pm 4.11$ |

Table 2: Comparison of performance (Accuracy in %) and calibration (ECE and ECE after Temperature Scaling), with Resnet101 on CIFAR10, CIFAR100 and Tiny-Imagenet datasets. †: Results reported from Noh et al. (2023). Best in **bold**, second best underlined, *for reported results and ours separately*.

| Methods | CIFAR10 | | | CIFAR100 | | | Tiny-Imagenet | | |
|---|---|---|---|---|---|---|---|---|---|
| | Acc. ($\uparrow$) | ECE ($\downarrow$) | TS ECE ($\downarrow$) | Acc. ($\uparrow$) | ECE ($\downarrow$) | TS ECE ($\downarrow$) | Acc. ($\uparrow$) | ECE ($\downarrow$) | TS ECE ($\downarrow$) |
| MMCE[†] | 94.99 | 3.88 | 1.15 | **77.82** | 13.43 | 3.06 | **66.44** | 3.40 | 3.40 |
| ECP[†] | 93.97 | 4.41 | 1.72 | 76.81 | 13.43 | 2.92 | $\underline{66.20}$ | 2.72 | 2.72 |
| LS[†] | 94.18 | 3.35 | 1.51 | 76.91 | 7.99 | 4.38 | 65.52 | 3.11 | 2.51 |
| FL[†] | 93.59 | $\underline{3.27}$ | $\underline{1.12}$ | 76.12 | **3.10** | $\underline{2.58}$ | 64.02 | $\underline{2.18}$ | 2.18 |
| FLSD[†] | 93.26 | 3.92 | **0.93** | 76.61 | $\underline{3.29}$ | **2.04** | 64.02 | **1.85** | 1.85 |
| CRL[†] | 95.04 | 3.74 | $\underline{1.12}$ | $\underline{77.60}$ | 7.29 | 3.32 | 65.87 | 3.57 | **1.60** |
| CPC[†] | **95.36** | 4.78 | 1.52 | 77.50 | 13.32 | 2.96 | **66.44** | 3.93 | 3.93 |
| MbLS[†] | $\underline{95.13}$ | **1.38** | 1.38 | 77.45 | 5.49 | 5.49 | 65.81 | $\underline{1.62}$ | 1.62 |
| ERM | $93.97 \pm 0.03$ | $4.30 \pm 0.01$ | $\underline{0.58 \pm 0.01}$ | $74.27 \pm 0.37$ | $14.33 \pm 0.17$ | $1.82 \pm 0.04$ | $65.7 \pm 0.07$ | $\underline{4.03 \pm 0.22}$ | $1.62 \pm 0.19$ |
| Mixup | $94.78 \pm 0.13$ | $2.20 \pm 0.23$ | $1.47 \pm 0.01$ | $77.50 \pm 0.18$ | $5.77 \pm 2.81$ | $2.58 \pm 0.01$ | $67.80 \pm 0.54$ | $5.49 \pm 1.11$ | $1.93 \pm 0.02$ |
| Manifold Mixup | $94.75 \pm 0.03$ | $\underline{2.13 \pm 0.12}$ | $1.57 \pm 0.05$ | $77.76 \pm 1.35$ | $5.34 \pm 0.04$ | $3.07 \pm 0.22$ | $68.57 \pm 0.04$ | $6.56 \pm 0.57$ | $2.11 \pm 0.01$ |
| RegMixup | $\underline{95.89 \pm 0.01}$ | $\mathbf{1.64 \pm 0.01}$ | $1.27 \pm 0.06$ | $\underline{78.85 \pm 0.08}$ | $6.72 \pm 0.08$ | $2.66 \pm 0.08$ | $\mathbf{69.76 \pm 0.07}$ | $10.19 \pm 0.06$ | $\mathbf{0.84 \pm 0.02}$ |
| RankMixup | $94.42 \pm 0.17$ | $2.87 \pm 0.25$ | $0.70 \pm 0.11$ | $77.27 \pm 0.08$ | $11.69 \pm 0.22$ | $2.12 \pm 0.20$ | $65.40 \pm 0.01$ | $13.85 \pm 9.60$ | $1.48 \pm 0.02$ |
| MIT-A | $95.27 \pm 0.01$ | $2.3 \pm 0.04$ | $0.93 \pm 0.01$ | $77.23 \pm 0.56$ | $9.55 \pm 0.46$ | $2.39 \pm 0.03$ | $68.03 \pm 0.09$ | $7.86 \pm 0.09$ | $1.52 \pm 0.15$ |
| SK Mixup *(Ours)* | $95.04 \pm 0.07$ | $2.20 \pm 0.12$ | $\mathbf{0.47 \pm 0.01}$ | $78.20 \pm 0.46$ | $\mathbf{2.23 \pm 0.39}$ | $\mathbf{1.11 \pm 0.06}$ | $67.60 \pm 0.01$ | $\mathbf{3.04 \pm 0.24}$ | $\underline{1.06 \pm 0.01}$ |
| SK RegMixup *(Ours)* | $\mathbf{96.02 \pm 0.04}$ | $2.36 \pm 0.59$ | $0.93 \pm 0.02$ | $\mathbf{79.57 \pm 0.03}$ | $\underline{4.32 \pm 1.21}$ | $\underline{1.54 \pm 0.10}$ | $\underline{69.11 \pm 0.06}$ | $7.19 \pm 0.38$ | $\underline{1.06 \pm 0.09}$ |

Error (UCE) (Laves et al., 2020) and Expected Normalized Calibration Error (ENCE) (Levi et al., 2022) for calibration. Results are reproduced and averaged over 10 different random runs. We also report standard deviation between the runs. A presentation of the different calibration metrics used can be found in Appendix C, along with a detailed description of implementation settings and hyperparameters in Appendix H.

## 4.2 CLASSIFICATION

**Effect of $\tau_{std}$** We present in Table 1 the performance and calibration results on CIFAR10 and CIFAR100 with a Resnet34, to study the effect of varying $\tau_{std}$. We compare the results with *Mixup* (Zhang et al., 2018) and *Mixup-IO* (Wang et al., 2023), which mixes between **I**nputs **O**nly while keeping one-hot labels, for varying values of $\alpha$. These two versions of Mixup show the trade-off between the effect of confidence penalty (in Mixup) that can hurt calibration on the one hand, and trivial confidence promotion (in Mixup-IO) that can hurt accuracy on the other hand. By varying only $\tau_{std}$, therefore changing the *extent* of mixing, we can achieve a better and finer trade-off between these two effects, and *improving both calibration and accuracy*. Furthermore, it shows that our SK Mixup obtains better results than simply changing $\alpha$ in Mixup and Mixup-IO. For other experiments, we first performed cross-validation to select $\tau_{std}$. In general, we found that $\tau_{std} = 0.25$ worked for all datasets and used by default $\tau_{max} = 1$ (unless stated otherwise). We refer the reader to Appendix J for more details on cross-validation.

**Comparison with state of the art** In Table 2, we present an extensive comparison of results on CIFAR10, CIFAR100 and Tiny-ImageNet with a Resnet101. We compare accuracy and calibration results reported *in the same settings* for various approaches from Noh et al. (2023), including *implicit*

Table 3: Comparison of performance (Accuracy in %) and calibration (ECE, AECE) with Resnet101 in Covariate-Shift, on CIFAR10-C and CIFAR100-C datasets, averaged over all corruptions and degrees of intensities. Best in **bold**, second best underlined.

| Methods | CIFAR10-C | | | CIFAR100-C | | |
|---|---|---|---|---|---|---|
| | Accuracy ($\uparrow$) | ECE ($\downarrow$) | AECE ($\downarrow$) | Accuracy ($\uparrow$) | ECE ($\downarrow$) | AECE ($\downarrow$) |
| ERM | $72.36 \pm 1.23$ | $21.80 \pm 0.55$ | $21.79 \pm 0.55$ | $47.96 \pm 0.59$ | $31.97 \pm 0.41$ | $31.96 \pm 0.41$ |
| Mixup | $77.46 \pm 1.09$ | $14.25 \pm 1.03$ | $14.22 \pm 1.01$ | $53.18 \pm 0.61$ | $17.73 \pm 4.07$ | $17.73 \pm 4.07$ |
| Manifold Mixup | $77.30 \pm 0.45$ | $14.14 \pm 3.38$ | $14.18 \pm 3.13$ | $53.42 \pm 1.8$ | $16.48 \pm 6.88$ | $16.48 \pm 6.88$ |
| MIT-A | $76.2 \pm 0.31$ | $15.46 \pm 1.0$ | $15.44 \pm 0.98$ | $53.36 \pm 0.27$ | $23.0 \pm 0.74$ | $23.0 \pm 0.74$ |
| RankMixup | $73.39 \pm 2.44$ | $19.16 \pm 2.45$ | $19.15 \pm 2.45$ | $47.44 \pm 2.36$ | $30.99 \pm 1.14$ | $30.99 \pm 1.14$ |
| RegMixup | $\underline{80.92 \pm 3.43}$ | $9.9 \pm 3.53$ | $9.8 \pm 3.45$ | $\underline{55.52 \pm 0.3}$ | $16.95 \pm 0.37$ | $16.95 \pm 0.37$ |
| SK Mixup (*Ours*) | $79.31 \pm 0.62$ | $\underline{6.87 \pm 0.94}$ | $\underline{6.89 \pm 0.94}$ | $55.04 \pm 1.28$ | $\underline{9.90 \pm 0.12}$ | $\underline{9.9 \pm 0.12}$ |
| SK RegMixup (*Ours*) | $\mathbf{81.69 \pm 2.9}$ | $\mathbf{5.88 \pm 1.53}$ | $\mathbf{5.9 \pm 1.56}$ | $\mathbf{57.98 \pm 0.32}$ | $\mathbf{6.89 \pm 1.38}$ | $\mathbf{6.9 \pm 1.37}$ |

Table 4: Comparison of In-Distribution (ID) performance (Accuracy in %) and calibration (ECE, AECE) on Imagenet dataset, and Out-of-distribution (OOD) performance and calibration on Imagenet-R and Imagenet-A datasets, with Resnet50 and ViT-S/16. Best in **bold**, second best underlined.

| Arch. | Methods | ID | | | OOD | | | | | |
|---|---|---|---|---|---|---|---|---|---|---|
| | | ImageNet | | | ImageNet-R | | | ImageNet-A | | |
| | | Acc. ($\uparrow$) | ECE ($\downarrow$) | AECE ($\downarrow$) | Acc. ($\uparrow$) | ECE ($\downarrow$) | AECE ($\downarrow$) | Acc. ($\uparrow$) | ECE ($\downarrow$) | AECE ($\downarrow$) |
| RN50 | ERM | 76.06 | 4.12 | 4.05 | 22.31 | 24.77 | 24.77 | 0.71 | 43 | 43 |
| | Mixup | 76.59 | 1.8 | 1.77 | 24.49 | 19.2 | 19.2 | 0.89 | 37.92 | 37.92 |
| | Manifold Mixup | 76.61 | $\underline{1.77}$ | 1.77 | 24.43 | 19.68 | 19.68 | 0.85 | 39.05 | 39.05 |
| | RegMixup | **77.82** | 2.61 | 2.59 | $\underline{25.70}$ | 22.67 | 22.67 | $\underline{2.27}$ | 40.14 | 40.14 |
| | RankMixup | 76.33 | 2.07 | 2.1 | 23.33 | 24.81 | 24.81 | 0.85 | 43.81 | 43.81 |
| | MIT-A | 76.7 | 1.8 | $\underline{1.74}$ | 24.86 | **17.83** | **17.83** | 0.81 | 37.37 | 37.37 |
| | SK Mixup (*Ours*) | 76.2 | **1.46** | **1.4** | 24.67 | 19.88 | 19.88 | 0.99 | $\underline{36.93}$ | $\underline{36.93}$ |
| | SK RegMixup (*Ours*) | $\underline{77.1}$ | 1.78 | 1.77 | **25.92** | $\underline{18.44}$ | $\underline{18.44}$ | **2.63** | **34.93** | **34.93** |
| ViT-S/16 | ERM | 69.34 | 9.03 | 9.03 | 15.46 | 35.93 | 35.93 | 1.83 | 45.03 | 45.03 |
| | Mixup | 72.0 | 5.81 | 5.8 | 18.21 | 29.29 | 29.29 | 2.47 | 41.07 | 41.07 |
| | Manifold Mixup | 72.04 | 6.99 | 6.95 | 18.93 | 30.27 | 30.27 | 2.15 | 43.47 | 43.47 |
| | RegMixup | $\underline{74.44}$ | 7.36 | 7.34 | $\underline{21.01}$ | 32.64 | 32.64 | $\underline{3.8}$ | 44.4 | 44.4 |
| | RankMixup | 69.99 | 9.30 | 9.30 | 15.77 | 37.37 | 37.37 | 1.77 | 46.82 | 46.82 |
| | MIT-A | 72.81 | 5.69 | 5.69 | 17.6 | 31.07 | 31.07 | 2.81 | 41.8 | 41.8 |
| | SK Mixup (*Ours*) | 71.83 | $\underline{3.89}$ | $\underline{3.9}$ | 18.37 | $\underline{28.56}$ | $\underline{28.56}$ | 2.28 | $\underline{39.33}$ | $\underline{39.33}$ |
| | SK RegMixup (*Ours*) | **75.11** | **2.0** | **1.98** | **22.06** | **26.23** | **26.23** | **4.44** | **38.25** | **38.25** |

*methods* such as LS (Müller et al., 2019), FL (Lin et al., 2017), FLSD (Mukhoti et al., 2020), MbLS (Liu et al., 2022a), and *explicit methods*, such as ECP (Pereyra et al., 2017), MMCE (Kumar et al., 2018), CRL (Moon et al., 2020), CPC (Cheng & Vasconcelos, 2022). We also compare results with Mixup, Manifold Mixup (Verma et al., 2019), RegMixup (Pinto et al., 2022), RankMixup (Noh et al., 2023) and MIT-A (Wang et al., 2023) in the same settings on multiple random seeds, using official codes and hyperparameters provided by the authors. We can see from these results that we achieve competitive accuracy and improves calibration compared to other *Calibration-driven Mixup methods*. Note that our approach always improves accuracy and calibration over ERM. We also show that our framework can be combined with RegMixup to further improve accuracy by about 1%. Then, we compare results in covariate shift settings in Table 3 against other calibration-driven Mixup methods. We observe that our SK Mixup achieves significantly better calibration results, about 3 points (of ECE and AECE) on CIFAR10-C and 7 points on CIFAR100-C that RegMixup, and that our SK RegMixup achieves both better accuracy and calibration than all other methods. In Table 4, we also evaluate the scalability of our method on ImageNet dataset, both for ID and OOD performance with ResNet and ViT models. While we achieve slightly lower ID accuracy than other method with our SK Mixup, we significantly improve ID calibration. Then, with SK RegMixup, we improve calibration over original RegMixup, with similar ID accuracy, and we achieve both better OOD accuracy and calibration than other methods.

Table 5: Performance (RMSE, MAPE) and calibration (UCE, ENCE) comparison on regression tasks. Best in **bold**, second best underlined.

| Methods | Airfoil MAPE (↓) | Airfoil UCE (↓) | Airfoil ENCE (↓) | Exchange Rate MAPE (↓) | Exchange Rate UCE (↓) | Exchange Rate ENCE (↓) | Electricity MAPE (↓) | Electricity UCE (↓) | Electricity ENCE (↓) |
|---|---|---|---|---|---|---|---|---|---|
| ERM | $1.720 \pm 0.219$ | $107.6 \pm 19.2$ | $2.10\% \pm 0.78$ | $1.924 \pm 0.287$ | $0.82\% \pm 0.28$ | $3.64\% \pm 0.74$ | $15.263 \pm 0.383$ | $0.76\% \pm 0.08$ | $\mathbf{22.07\% \pm 1.27}$ |
| Mixup | $2.003 \pm 0.126$ | $147.1 \pm 34.0$ | $2.12\% \pm 0.63$ | $1.926 \pm 0.284$ | $\mathbf{0.74\% \pm 0.22}$ | $\underline{3.52\% \pm 0.59}$ | $14.944 \pm 0.386$ | $0.62\% \pm 0.07$ | $24.28\% \pm 2.47$ |
| Manifold Mixup | $1.964 \pm 0.111$ | $126.0 \pm 15.8$ | $2.06\% \pm 0.64$ | $2.006 \pm 0.346$ | $0.86\% \pm 0.29$ | $3.82\% \pm 0.85$ | $14.872 \pm 0.409$ | $0.69\% \pm 0.05$ | $24.53\% \pm 1.44$ |
| RegMixup | $1.725 \pm 0.092$ | $\underline{105.0 \pm 23.1}$ | $1.92\% \pm 0.73$ | $1.918 \pm 0.290$ | $0.80\% \pm 0.24$ | $3.66\% \pm 0.73$ | $\underline{14.790 \pm 0.395}$ | $0.66\% \pm 0.12$ | $23.68\% \pm 2.53$ |
| C-Mixup | $\underline{1.706 \pm 0.104}$ | $111.2 \pm 32.6$ | $\underline{1.90\% \pm 0.75}$ | $\underline{1.893 \pm 0.222}$ | $0.78\% \pm 0.20$ | $3.60\% \pm 0.64$ | $15.085 \pm 0.533$ | $0.69\% \pm 0.11$ | $\underline{23.54\% \pm 1.90}$ |
| SK Mixup (Ours) | $\mathbf{1.609 \pm 0.137}$ | $\mathbf{93.8 \pm 25.5}$ | $\mathbf{1.85\% \pm 0.84}$ | $\mathbf{1.814 \pm 0.241}$ | $\underline{0.76\% \pm 0.21}$ | $3.40\% \pm 0.56$ | $\mathbf{14.649 \pm 0.191}$ | $\mathbf{0.61\% \pm 0.05}$ | $23.93\% \pm 2.05$ |

Table 6: Efficiency comparison between Mixup methods. We report the number of batch of data in memory at each iteration, the best epoch measured on validation data, time per epoch (in seconds) and total training time to reach the best epoch (in seconds), for Image classification on CIFAR10, CIFAR100 and Tiny-imagenet datasets, with a Resnet50.

| Method | Batch in memory | CIFAR10 Best Epoch | CIFAR10 Time | CIFAR10 Total Time | CIFAR100 Best Epoch | CIFAR100 Time | CIFAR100 Total Time | Tiny-Imagenet Best Epoch | Tiny-Imagenet Time | Tiny-Imagenet Total Time |
|---|---|---|---|---|---|---|---|---|---|---|
| Mixup | **1** | 186 | **17** | **3162** | 181 | **17** | **3077** | 89 | **125** | **11125** |
| RankMixup | 4 | **147** | 78 | 11485 | 177 | 78 | 13806 | 80 | 540 | 43380 |
| MIT-A | 2 | 189 | 46 | 8694 | $\underline{167}$ | 46 | 7659 | 82 | 305 | 24908 |
| RegMixup | 2 | $\underline{173}$ | 32 | 5536 | 177 | 32 | 5664 | 94 | 240 | 22560 |
| SK Mixup | **1** | 183 | $\underline{28}$ | $\underline{5138}$ | **160** | $\underline{28}$ | $\underline{4480}$ | **69** | $\underline{189}$ | $\underline{13104}$ |
| SK RegMixup | 2 | 175 | 39 | 6825 | 179 | 39 | 6981 | $\underline{75}$ | 316 | 23700 |

## 4.3 REGRESSION

To demonstrate the flexibility of our framework regarding different tasks, we provide experiments on regression for tabular data and time series. Regression tasks have the advantage of having an obvious meaningful distance between points, which is distance between labels. Therefore, following Yao et al. (2022a), we directly measure the similarity between two points by the *distance between their labels*, *i.e.*, $\bar{d}_n(y_i, y_{\sigma(i)})$. This avoids the computation of distance on embeddings and makes our method *as fast as the original Mixup*. In Table 5, we compare our SK Mixup with Mixup (Zhang et al., 2018), Manifold Mixup (Verma et al., 2019), RegMixup (Pinto et al., 2022), and C-Mixup (Yao et al., 2022a). We can see that our approach achieves competitive results with state-of-the-art C-Mixup, in both performance (MAPE) and calibration metrics.

## 4.4 EFFICIENCY COMPARISON

Our method achieves competitive results while being *much more efficient* than other state-of-the-art approaches. Indeed, as can be seen in Table 6, our SK Mixup is about $1.5\times$ *faster* than MIT-A, about $3\times$ faster than RankMixup while using a single batch of data like Mixup. We present a full comparison in Appendix N.

## 5 CONCLUSION

Motivated by calibration in both classification and regression tasks, we present *Similarity Kernel Mixup*, a flexible framework to take into account distance between data for linearly interpolation during training, based on a *similarity kernel*. The coefficients governing the interpolation are warped to change their underlying distribution depending on the similarity between the points to mix, such that similar data are mixed more strongly than less similar ones, preserving calibration by avoiding *manifold mismatch*, as we prove that likelihood of assigning noisy label when mixing increases with distance. This provides a *more efficient* data augmentation approach than *Calibration-driven Mixup methods*, both in terms of time and memory, improving both accuracy and calibration, even more out-of-distribution. We illustrate through extensive experiments the effectiveness of the approach in classification as well as in regression, spanning multiple neural network architectures including CNNs, ViTs, MLPs and RNNs. We also show that our proposed framework can be combined with RegMixup (Pinto et al., 2022) to further boost performance. Future works include applications to more complex tasks such as semantic segmentation, depth estimation, or structured data.

## REPRODUCIBILITY STATEMENT

Throughout the paper, we made sure that all our experiments were fully reproducible, describing in details all datasets and architectures considered in Section 4.1, common evaluation metrics for calibration in Appendix C, and all hyperparameters and training settings in Appendix H. We also describe how cross validation and the selection of hyperparameters are performed in Appendix J.

## ACKNOWLEDGEMENTS

This work was supported by the PEPR IA FOUNDRY project (ANR-23-PEIA-0003) of the French National Research Agency (ANR). We also thank all anonymous reviewers for providing constructive comments and Andrei Bursuc for providing feedback on an early version of the manuscript.

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

## A   Broader Impact

This paper presents work whose goal is to improve the calibration of machine learning models. Calibration is the process of improving the reliability of the confidence score associated with predictions. This is an important step towards the development of trustworthy models. Having better calibrated models can also help in the detection of biases, overfitting or out-of-distribution data, which can prevent models from being misused in practice.

## B   Limitations

The majority of theoretical analysis around mixup have been focused on generalization, while we are interested on calibration. Analyzing the impact on calibration is very difficult, as we also need to take into account the effect of Temperature Scaling (TS) (Guo et al., 2017) on a validation set. Indeed, TS can change ordering of results (Ashukha et al., 2020), and, more importantly, Wang et al. (2023) observed that calibration using Mixup can be worse than ERM after TS. This is an observation that we also confirm in our experiments (e.g. in Table 1). Thus, an analysis including temperature scaling would be required first for vanilla Mixup before analyzing our proposed method. This would be a significant contribution on its own, and our future works include exploring this direction.

Our approach also introduces two additional hyperparameters. However, since in our case we always draw initial parameters $\lambda$ from $\texttt{Beta}(1, 1)$, $\alpha$ is not a hyperparameter anymore. Then, $\tau_{max}$ and $\tau_{std}$ can be tuned separately, and, as mentioned in Section 4.2, we found that impact on calibration was mainly controlled by $\tau_{std}$. Furthermore, one should note that we always used $\tau_{std} = 0.25$ in image classification experiments, showing our approach is not sensitive to hyperparameter choice between datasets. We discuss selection of hyperparameters with cross-validation in Appendix J.

## C  INTRODUCTION TO CALIBRATION METRICS

As discussed in Section 2.2, calibration measures the difference between predictive confidence and actual probability. More formally, with $\hat{y}$ and $y \in \mathbb{Y}$, respectively the model's prediction and target label, and $\hat{p}$ its predicted confidence, a perfectly calibrated model should satisfy $P(\hat{y} = y | \hat{p} = p) = p$, for $p \in [0, 1]$.

We use several metrics for calibration in the paper, namely, ECE, AECE, Brier score and NLL for classification tasks, and UCE and ENCE for regression tasks. We formally introduce all of them here.

### C.1  METRICS FOR CLASSIFICATION TASKS

**NLL**  The *negative log-likelihood* (NLL) is a common metric for a model's prediction quality (Hastie et al., 2009). It is equivalent to cross-entropy in multi-class classification. NLL is defined as:

$$\text{NLL}(\mathbf{x}, \mathbf{y}) = -\frac{1}{N} \sum_{i=1}^{N} \log(\hat{p}(\mathbf{y}_i | \mathbf{x}_i)), \tag{6}$$

where $\hat{p}(\mathbf{y}_i | \mathbf{x}_i)$ represents the confidence of the model in the output associated to $\mathbf{x}_i$ for the target class $\mathbf{y}_i$.

**Brier score**  The Brier score (Brier, 1950) for multi-class classification is defined as

$$\text{Brier}(\mathbf{x}, \mathbf{y}) = -\frac{1}{N} \sum_{i=1}^{N} \sum_{j=1}^{c} (\hat{p}(y_{(i,j)} | \mathbf{x}_i) - y_{(i,j)})^2, \tag{7}$$

where we assume that the target label $\mathbf{y}_i$ is represented as a one-hot vector over the $c$ possible class, *i.e.*, $\mathbf{y}_i \in \mathbb{R}^c$. Brier score is the mean square error (MSE) between predicted confidence and target.

**ECE**  Expected Calibration Error (ECE) is a popular metric for calibration performance for classification tasks in practice. It approximates the difference between accuracy and confidence in expectation by first grouping all the samples into $M$ equally spaced bins $\{B_m\}_{m=1}^{M}$ with respect to their confidence scores, then taking a weighted average of the difference between accuracy and confidence for each bin. Formally, ECE is defined as (Guo et al., 2017):

$$\text{ECE} := \sum_{m=1}^{M} \frac{|B_m|}{N} |\text{acc}(B_m) - \text{conf}(B_m)|, \tag{8}$$

with $\text{acc}(B_m) = \frac{1}{|B_m|} \sum_{i \in B_m} \mathbb{1}_{\hat{y}_i = y_i}$ the accuracy of bin $B_m$, and $\text{conf}(B_m) = \frac{1}{|B_m|} \sum_{i \in B_m} \hat{p}(\mathbf{y}_i | \mathbf{x}_i)$ the average confidence within bin $B_m$.

**AECE**  The Adaptive ECE (AECE) is computed similarly to ECE, with the difference that bin sizes are calculated to evenly distribute samples across the bins.

### C.2  METRICS FOR REGRESSION TASKS

A probabilistic regression model takes $\mathbf{x} \in \mathbb{X}$ as input and outputs a mean $\mu_y(\mathbf{x})$ and a variance $\sigma_y^2(\mathbf{x})$ targeting the ground-truth $y \in \mathbb{Y}$. The UCE and ENCE calibration metrics are both extension of ECE for regression tasks to evaluate *variance calibration*. They both apply a binning scheme with $M$ bins over the predicted variance.

**UCE**  Uncertainty Calibration Error (UCE) (Laves et al., 2020) measures the average of the absolute difference between *mean squared error (MSE)* and *mean variance (MV)* within each bin. It is formally defined by

$$\text{UCE} := \sum_{m=1}^{M} \frac{|B_m|}{N} |\text{MSE}(B_m) - \text{MV}(B_m)|, \tag{9}$$

with $\text{MSE}(B_m) = \frac{1}{|B_m|} \sum_{i \in B_m} (\mu_{y_i}(\mathbf{x}_i) - y_i)^2$ and $\text{MV}(B_m) = \frac{1}{|B_m|} \sum_{i \in B_m} \sigma_{y_i}^2(\mathbf{x}_i)^2$.

**ENCE** Expected Normalized Calibration Error (ENCE) (Levi et al., 2022) measures the absolute *normalized* difference, between *root mean squared error (RMSE)* and *root mean variance (RMV)* within each bin. It is formally defined by

$$\text{ENCE} := \frac{1}{M} \sum_{m=1}^{M} \frac{|\text{RMSE}(B_m) - \text{RMV}(B_m)|}{\text{RMV}(B_m)}, \tag{10}$$

with $\text{RMSE}(B_m) = \sqrt{\frac{1}{|B_m|} \sum_{i \in B_m} (\mu_{y_i}(\mathbf{x}_i) - y_i)^2}$ and $\text{RMV}(B_m) = \sqrt{\frac{1}{|B_m|} \sum_{i \in B_m} \sigma_{y_i}^2(\mathbf{x}_i)^2}$.

Table 7: Performance (Accuracy in %) and calibration (ECE, Brier, NLL) *before and after Temperature Scaling (TS)* (Guo et al., 2017) with Resnet34 when mixing only elements higher or lower than a quantile $q$. Best in **bold**, second best underlined.

| Dataset | Quantile of Distance | Accuracy ($\uparrow$) | ECE ($\downarrow$) | Brier ($\downarrow$) | NLL ($\downarrow$) | TS ECE ($\downarrow$) | TS Brier ($\downarrow$) | TS NLL ($\downarrow$) |
|---|---|---|---|---|---|---|---|---|
| | Lower 0.0 / Higher 1.0 (ERM) | $94.69 \pm 0.27$ | $3.65 \pm 0.22$ | $8.90 \pm 0.4$ | $24.57 \pm 1.04$ | **$0.82 \pm 0.11$** | $8.07 \pm 0.31$ | $17.50 \pm 0.61$ |
| | Lower 1.0 / Higher 0.0 (Mixup) | $95.97 \pm 0.27$ | $12.3 \pm 0.92$ | $8.54 \pm 0.64$ | $25.68 \pm 1.53$ | $1.36 \pm 0.13$ | $6.53 \pm 0.36$ | $16.35 \pm 0.72$ |
| | Lower 0.1 | $95.70 \pm 0.24$ | **$1.58 \pm 0.3$** | $7.12 \pm 0.16$ | $17.9 \pm 1.01$ | $0.99 \pm 0.3$ | $7.08 \pm 0.37$ | $16.1 \pm 0.85$ |
| | Lower 0.25 | $95.73 \pm 0.18$ | $\underline{2.42 \pm 0.55}$ | $7.12 \pm 0.25$ | $19.54 \pm 1.1$ | $1.74 \pm 0.45$ | $7.07 \pm 0.26$ | $19.39 \pm 1.11$ |
| C10 | Lower 0.5 | $95.88 \pm 0.28$ | $3.04 \pm 0.4$ | $6.67 \pm 0.34$ | $16.68 \pm 0.84$ | $1.56 \pm 0.28$ | $6.68 \pm 0.34$ | $15.86 \pm 0.71$ |
| | Lower 0.75 | $96.16 \pm 0.09$ | $3.63 \pm 0.32$ | $6.33 \pm 0.14$ | **$16.55 \pm 0.59$** | $1.12 \pm 0.16$ | $6.35 \pm 0.15$ | $\underline{15.20 \pm 0.44}$ |
| | Lower 0.9 | **$96.31 \pm 0.08$** | $3.71 \pm 0.34$ | $\underline{6.20 \pm 0.24}$ | $16.65 \pm 0.41$ | $1.10 \pm 0.05$ | **$6.14 \pm 0.11$** | **$15.16 \pm 0.29$** |
| | Higher 0.9 | $95.58 \pm 0.34$ | $2.72 \pm 0.2$ | $7.39 \pm 0.49$ | $20.54 \pm 1.31$ | $1.86 \pm 0.25$ | $7.4 \pm 0.48$ | $20.32 \pm 1.25$ |
| | Higher 0.75 | $95.91 \pm 0.14$ | $3.68 \pm 0.19$ | $6.86 \pm 0.21$ | $20.7 \pm 0.88$ | $1.85 \pm 0.17$ | $6.84 \pm 0.22$ | $20.06 \pm 1.12$ |
| | Higher 0.5 | $95.58 \pm 0.28$ | $4.55 \pm 0.63$ | $7.28 \pm 0.38$ | $20.71 \pm 1.14$ | $1.67 \pm 0.13$ | $7.23 \pm 0.37$ | $19.12 \pm 0.74$ |
| | Higher 0.25 | $95.98 \pm 0.3$ | $4.28 \pm 0.32$ | $6.66 \pm 0.49$ | $18.73 \pm 0.97$ | $1.24 \pm 0.18$ | $6.65 \pm 0.51$ | $17.06 \pm 0.99$ |
| | Higher 0.1 | $\underline{96.28 \pm 0.03}$ | $4.38 \pm 0.16$ | **$6.15 \pm 0.05$** | $17.18 \pm 0.28$ | $1.13 \pm 0.11$ | **$6.14 \pm 0.04$** | $15.24 \pm 0.37$ |
| | Lower 0.0 / Higher 1.0 (ERM) | $73.47 \pm 1.59$ | $13.0 \pm 0.75$ | $39.56 \pm 2.19$ | $121.06 \pm 7.92$ | $2.54 \pm 0.15$ | $36.47 \pm 2.05$ | $100.82 \pm 6.93$ |
| | Lower 1.0 / Higher 0.0 (Mixup) | $78.11 \pm 0.57$ | $11.92 \pm 0.73$ | $32.82 \pm 0.87$ | $95.76 \pm 2.43$ | $2.49 \pm 0.19$ | $31.06 \pm 0.69$ | $87.94 \pm 1.98$ |
| | Lower 0.1 | $75.40 \pm 0.53$ | $4.72 \pm 0.32$ | $35.87 \pm 0.52$ | $108.38 \pm 1.12$ | $3.48 \pm 0.24$ | $35.92 \pm 0.5$ | $105.87 \pm 1.41$ |
| | Lower 0.25 | $77.14 \pm 0.51$ | **$2.56 \pm 0.17$** | $32.93 \pm 0.64$ | $95.47 \pm 1.75$ | $2.54 \pm 0.22$ | $32.95 \pm 0.62$ | $95.42 \pm 1.78$ |
| C100 | Lower 0.5 | $77.66 \pm 0.15$ | $3.40 \pm 1.17$ | $32.10 \pm 0.41$ | $90.75 \pm 2.02$ | **$1.85 \pm 0.43$** | $31.94 \pm 0.28$ | $89.97 \pm 1.53$ |
| | Lower 0.75 | $78.43 \pm 0.62$ | $4.38 \pm 1.58$ | $30.85 \pm 0.65$ | $86.83 \pm 1.81$ | $1.95 \pm 0.6$ | $30.64 \pm 0.7$ | $85.06 \pm 1.89$ |
| | Lower 0.9 | **$79.24 \pm 0.7$** | $6.02 \pm 1.02$ | $\underline{30.11 \pm 1.05}$ | $85.53 \pm 3.32$ | $1.99 \pm 0.03$ | $\underline{29.72 \pm 0.94}$ | $\underline{82.54 \pm 2.82}$ |
| | Higher 0.9 | $77.3 \pm 0.43$ | $3.55 \pm 1.66$ | $32.19 \pm 0.78$ | $90.54 \pm 2.56$ | $\underline{1.92 \pm 0.22}$ | $32.0 \pm 0.59$ | $88.69 \pm 1.67$ |
| | Higher 0.75 | $77.8 \pm 1.05$ | $3.77 \pm 1.14$ | $31.56 \pm 1.23$ | $90.16 \pm 4.16$ | $2.29 \pm 0.24$ | $31.48 \pm 1.18$ | $88.16 \pm 3.86$ |
| | Higher 0.5 | $78.74 \pm 0.43$ | $4.32 \pm 0.96$ | $30.47 \pm 0.54$ | $86.96 \pm 1.87$ | $2.52 \pm 0.22$ | $30.37 \pm 0.56$ | $84.64 \pm 1.63$ |
| | Higher 0.25 | $78.51 \pm 0.47$ | $\underline{3.35 \pm 1.19}$ | $30.13 \pm 0.6$ | $\underline{85.49 \pm 2.6}$ | $2.34 \pm 0.26$ | $30.42 \pm 0.59$ | $84.64 \pm 2.23$ |
| | Higher 0.1 | $\underline{79.14 \pm 0.53}$ | $5.32 \pm 2.15$ | **$29.94 \pm 0.76$** | $85.09 \pm 2.53$ | $2.23 \pm 0.34$ | **$29.62 \pm 0.51$** | **$82.22 \pm 1.28$** |

## D EFFECT OF DISTANCE ON CALIBRATION

In Table 7, we show the exact results for the plot presented in Section 3.1, along with results obtained for different quantiles $q$. One should compare results of "Lower $q$" with "Higher $1 - q$" to have equivalent numbers of possible element to mix with (*diversity*). We repeat our observations here for ease of reading. First, we observe that Mixup improves upon ERM's accuracy, but can degrade calibration depending on the dataset, which is consistent with findings from Wang et al. (2023). Then, when selecting pairs according to distance, a sufficiently high proportion of data to mix is necessary to preserve accuracy ($q > 0.5$). Finally, mixing data with *lower* distances achieves a better calibration as opposed to mixing data with *higher* distances.

## E DETAILED ALGORITHM

---

**Algorithm 1** Similarity Kernel Mixup training procedure

---

**Input:** Batch of data $\mathcal{B} = \{(\mathbf{x}_i, y_i)\}_{i=1}^{n}$, similarity parameters $(\tau_{\max}, \tau_{\text{std}})$, model parameters at the current iteration $\theta_t$
$\tilde{\mathcal{B}} \leftarrow \varnothing$
$\sigma_t \sim \mathfrak{S}_n$          {Sample random permutation}
**for** $\forall i \in \{1, \ldots, n\}$ **do**
    $\lambda_i \sim \texttt{Beta}(1, 1)$
    $\tau_i := \tau(\mathbf{x}, i, \sigma; \tau_{\max}, \tau_{\text{std}})$     {Compute warping parameters through Equations (4) and (5)}
    $\tilde{\mathbf{x}}_i := \omega_{\tau_i}(\lambda_i)\mathbf{x}_i + (1 - \omega_{\tau_i}(\lambda_i))\mathbf{x}_{\sigma(i)}$      {Generate new data}
    $\tilde{y}_i := \omega_{\tau_i}(\lambda_i)y_i + (1 - \omega_{\tau_i}(\lambda_i))y_{\sigma(i)}$      {Generate new labels}
    $\tilde{\mathcal{B}} \leftarrow \tilde{\mathcal{B}} \cup (\tilde{\mathbf{x}}_i, \tilde{y}_i)$      {Aggregate new batch}
**end for**
Compute and optimize loss over $\tilde{\mathcal{B}}$
**Output:** updated parameters of the model $\theta_{t+1}$

---

We present a pseudocode of our *Similarity Kernel Mixup* procedure for a single training iteration in Algorithm 1. The generation of new data is explained in the pseudocode as a sequential process for simplicity and ease of understanding, but the actual implementation is optimized to work in parallel on GPU through vectorized operations.

# F  PROOFS

## F.1  THEOREM 3.1

We give the full proof of Theorem 3.1 below.

### F.1.1  PROOF OF THE FIRST PART

*Proof.* For (i), we will separate the cases in two depending on the intersection of the convex hulls of the two manifolds $\bar{\mathcal{M}}_i \cap \bar{\mathcal{M}}_j$ being empty or not.

**Intersection not empty**  Let us first suppose that $\bar{\mathcal{M}}_i \cap \bar{\mathcal{M}}_j \neq \emptyset$ and take $\mathbf{z}$ from $\bar{\mathcal{M}}_i \cap \bar{\mathcal{M}}_j$. Then, since the manifolds are disjoint, $\mathbf{z}$ belongs either to $\mathcal{M}_i$, $\mathcal{M}_j$ or none of the two.

Suppose that $\mathbf{z} \in \mathcal{M}_i$. Since we also have $\mathbf{z} \in \bar{\mathcal{M}}_j$, there exists $\mathbf{x}_k, \mathbf{x}_l \in \mathcal{M}_j$ and $\lambda_0 \in [0, 1]$, such that $\mathbf{z} = \lambda_0 \mathbf{x}_k + (1 - \lambda_0)\mathbf{x}_l$, thanks to convexity. On the line segment $[\mathbf{x}_k, \mathbf{x}_l]$, since $\mathbf{z} \in \mathcal{M}_i$ and manifolds are disjoint , $\exists (\lambda_1, \lambda_2) \in [0, 1]^2, \lambda_1 > \lambda_2$ such that $\forall \lambda \in ]\lambda_2, \lambda_1[, \tilde{\mathbf{x}}(\lambda) = \lambda \mathbf{x}_k + (1 - \lambda)\mathbf{x}_l \notin \mathcal{M}_j$. Then, $\lambda_0 \in ]\lambda_2, \lambda_1[$.

With the same reasoning, if $\mathbf{z} \in \mathcal{M}_j$, we can find $\mathbf{x}_k, \mathbf{x}_l \in \mathcal{M}_i$. Finally, if $\mathbf{z}$ belongs neither in $\mathcal{M}_i$ nor $\mathcal{M}_j$, then we can find $\mathbf{x}_k, \mathbf{x}_l$ both in either manifold since $\mathbf{z} \in \bar{\mathcal{M}}_i \cap \bar{\mathcal{M}}_j$ and obtain the same result.

**Empty intersection**  Now, suppose that $\bar{\mathcal{M}}_i \cap \bar{\mathcal{M}}_j = \emptyset$. Without loss of generality, we can consider $\mathbf{x}_k \in \mathcal{M}_i$ and $\mathbf{x}_l \in \mathcal{M}_j$. Then, $\forall \lambda \in [0, 1]$, we define $\tilde{\mathbf{x}}(\lambda) = \lambda \mathbf{x}_k + (1 - \lambda)\mathbf{x}_j$, the linear convex combination of the two points weighted by $\lambda$. Then, since the convex hulls of the manifolds are disjoints, there exists $\lambda_1 \in [0, 1]$, such that $\forall \lambda \geq \lambda_1, \tilde{\mathbf{x}}(\lambda) \in \bar{\mathcal{M}}_i$ and $\forall \lambda < \lambda_1, \tilde{\mathbf{x}}(\lambda) \notin \bar{\mathcal{M}}_i$. Symmetrically, there exists $\lambda_2 \in [0, 1]$, such that $\forall \lambda \leq \lambda_2, \tilde{\mathbf{x}}(\lambda) \in \bar{\mathcal{M}}_j$, and $\forall \lambda > \lambda_2, \tilde{\mathbf{x}}(\lambda) \notin \bar{\mathcal{M}}_j$. The convex hulls of the manifolds being disjoints, we have that $\lambda_1 > \lambda_2$.  □

### F.1.2  PROOF OF THE SECOND PART

*Proof.* Result (ii) is obtained directly by rewriting:

$$\|\tilde{\mathbf{x}}(\lambda_1) - \tilde{\mathbf{x}}(\lambda_2)\|_{\mathcal{H}} = \|\lambda_1 \mathbf{x}_k + (1 - \lambda_1)\mathbf{x}_l - (\lambda_2 \mathbf{x}_k + (1 - \lambda_2)\mathbf{x}_l)\|_{\mathcal{H}} \tag{11}$$

$$= \|\mathbf{x}_l + \lambda_1(\mathbf{x}_k - \mathbf{x}_l) - \mathbf{x}_l - \lambda_2(\mathbf{x}_k - \mathbf{x}_l)\|_{\mathcal{H}} \tag{12}$$

$$= |\lambda_1 - \lambda_2| \|\mathbf{x}_k - \mathbf{x}_l\|_{\mathcal{H}} \tag{13}$$

□

## F.2  THEOREM 3.2

We start by defining the *Total Variation* and the *Wasserstein metric*, and necessary lemmas.

**Definition F.1.** (Total Variation) The total variation between two probability measures $\mu$ and $\nu$ on $\mathbb{X}$ is

$$TV(\mu, \nu) := \sup_{E \subset \mathbb{X}} |\mu(E) - \nu(E)|, \tag{14}$$

where the supremum is taken over all measurable sets $E \subset \mathbb{X}$.

**Lemma F.2.** *(Levin & Peres (2017), Proposition 4.2) Let $\mu$ and $\nu$ be two probability measures on $\mathbb{X}$. If $\mathbb{X}$ is countable, then*

$$TV(\mu, \nu) = \frac{1}{2} \sum_{\mathbf{x} \in \mathbb{X}} |\mu(\mathbf{x}) - \nu(\mathbf{x})|. \tag{15}$$

**Lemma F.3.** *(Coupling Inequality, Levin & Peres (2017), Proposition 4.7) Let $\mu$ and $\nu$ be two probability measures on $\mathbb{X}$. Then any coupling of random variables $(X, Y)$ with respective marginals $\mu$ and $\nu$ satisfies*

$$TV(\mu, \nu) \leq P(X \neq Y). \tag{16}$$

**Definition F.4.** (Wasserstein metric) The Wasserstein metric between two probability measures $\mu$ and $\nu$ on $\mathbb{X}$ is

$$W_1(\mu, \nu) := \inf_{\pi \in \Pi(\mu, \nu)} \int_{\mathbb{X} \times \mathbb{X}} \|\mathbf{x} - \mathbf{x}'\|_{\mathcal{H}} d\pi(\mathbf{x}, \mathbf{x}'), \tag{17}$$

where the infimum is taken over $\Pi(\mu, \nu)$, defined as the set of all probability measures $\pi$ on $\mathbb{X} \times \mathbb{X}$ with respective marginals $\mu$ and $\nu$.

**Lemma F.5.** *(Gibbs & Su (2002), Theorem 4) Let $\mu$ and $\nu$ be two probability measures on $\mathbb{X}$. Then, the Wasserstein metric and the Total Variation distance satisfy the following relation,*

$$W_1(\mu, \nu) \leq R \cdot TV(\mu, \nu), \tag{18}$$

*where $R = \sup_{\mathbf{x}, \mathbf{x}' \in \mathbb{X}} \|\mathbf{x} - \mathbf{x}'\|_{\mathcal{H}}$ is the radius of the space.*

We can now prove Theorem 3.2:

*Proof.* By Lemma F.3, we have

$$P(\tilde{Y}^* \neq \tilde{Y}|\tilde{X}) \geq TV(P(\tilde{Y}|\tilde{X}), P(\tilde{Y}^*|\tilde{X}))$$

However, we also have

$$TV(P(\tilde{Y}|\tilde{X}), P(\tilde{Y}^*|\tilde{X})) = TV(P(\tilde{Y}|\tilde{X}), P(Y|X)) = TV(P(\tilde{Y}|\tilde{X}), P(Y'|X')).$$

Then, from Lemma F.2, we obtain

$$\begin{aligned}
TV(P(\tilde{Y}|\tilde{X}), P(Y|X)) &= \frac{1}{2} \sum_{m=1}^{M} |P(Y = m|X) - P(\tilde{Y} = m|\tilde{X})| \\
&= \frac{1}{2} \sum_{m=1}^{M} |f_m(X) - ((1-\lambda)f_m(X) + \lambda f_m(X'))| \\
&= \lambda TV(P(Y|X), P(Y'|X')).
\end{aligned}$$

And symmetrically,

$$\begin{aligned}
TV(P(\tilde{Y}|\tilde{X}), P(Y'|X')) &= \frac{1}{2} \sum_{m=1}^{M} |P(Y' = m|X') - P(\tilde{Y} = m|\tilde{X})| \\
&= \frac{1}{2} \sum_{m=1}^{M} |f_m(X') - ((1-\lambda)f_m(X) + \lambda f_m(X'))| \\
&= (1-\lambda) TV(P(Y|X), P(Y'|X')),
\end{aligned}$$

which both gives us our first inequality:

$$P(\tilde{Y}^* \neq \tilde{Y}|\tilde{X}) \geq \min(\lambda, (1-\lambda)) TV(P(Y|X), P(Y'|X')).$$

Now, we come back to our classification problem with $M$ classes, where samples $(\mathbf{x}, y) \in \mathbb{X} \times \{1, \ldots, M\}$ are drawn from probability measures $\nu_y$ respectively supported on the class manifolds $\mathcal{M}_y$. Then, $TV(P(Y|X = \mathbf{x}), P(Y'|X' = \mathbf{x}')) = TV(\mathcal{M}_y, \mathcal{M}_{y'})$, and using Lemma F.5 concludes the proof. $\square$

### F.3 PROPOSITION 3.2

We give the proof of Proposition 3.3 below:

*Proof.* Let $\tau > 0$ and $F_\tau(x) = I_x(\tau, \tau)$, then $F_\tau$ is the *cumulative distribution function (CDF)* of a Beta distribution with parameters $(\tau, \tau)$. $\forall x \in [0, 1]$, we have:

$$P(\omega_\tau(\lambda) \leq x) = P(I_\lambda^{-1}(\tau, \tau) \leq x) \tag{19}$$

$$= P(\lambda \leq I_x(\tau, \tau)) \tag{20}$$

$$= I_x(\tau, \tau), \tag{21}$$

where Equation (20) is obtained since $F_\tau$ is continuous and monotonically increasing on $[0, 1]$, and Equation (21) because $\lambda$ follows a uniform distribution on $[0, 1]$. It follows that $\omega_\tau(\lambda) \sim \texttt{Beta}(\tau, \tau)$. □

Table 8: Comparison of performance (Accuracy in %), calibration (ECE, AECE) after Temperature Scaling, and total training time (in min), with three different variants. All experiments are using a ResNet34 on CIFAR10 on a single A100 GPU and reproduced on 4 different random seeds.

| Method | Accuracy ($\uparrow$) | ECE ($\downarrow$) | AECE ($\downarrow$) | Total training time (in min) ($\downarrow$) |
|---|---|---|---|---|
| Warping | $\mathbf{96.42 \pm 0.24}$ | $\mathbf{0.53 \pm 0.06}$ | $0.71 \pm 0.05$ | $57.36 \pm 0.31$ |
| Warping w/ lookup | $96.04 \pm 0.2$ | $0.57 \pm 0.1$ | $\mathbf{0.70 \pm 0.12}$ | $\mathbf{55.59 \pm 0.4}$ |
| No warping | $96.3 \pm 0.1$ | $0.55 \pm 0.07$ | $0.76 \pm 0.16$ | $56.88 \pm 1.03$ |

## G  BENEFITS OF WARPING

### G.1  DISENTANGLING INPUTS AND TARGETS

Using such warping functions presents the advantage of being able to easily separate the mixing of *inputs* and *targets*, by defining different *warping parameters* $\tau^{(i)}$ and $\tau^{(t)}$:

$$\lambda_t \sim \texttt{Beta}(1,1), \qquad \begin{aligned} \tilde{\mathbf{x}}_i &:= \omega_{\tau^{(i)}}(\lambda_t)\mathbf{x}_i + (1 - \omega_{\tau^{(i)}}(\lambda_t))\mathbf{x}_{\sigma_t(i)} \\ \tilde{y}_i &:= \omega_{\tau^{(t)}}(\lambda_t)y_i + (1 - \omega_{\tau^{(t)}}(\lambda_t))y_{\sigma_t(i)}. \end{aligned} \tag{22}$$

Disentangling *inputs* and *targets* can be interesting when working in the imbalanced setting (Chou et al., 2020). Notably, with $\tau^{(i)} = 1, \tau^{(t)} \approx 0$, we recover the Mixup Input-Only (IO) variant (Wang et al., 2023) where only inputs are mixed, used in the experiments in Table 1, and with $\tau^{(i)} \approx 0, \tau^{(t)} = 1$, the Mixup Target-Only (TO) variant (Wang et al., 2023), where only labels are mixed. It can also reveal interesting for *structured prediction*, where targets are structured objects such as *graph prediction* or *depth estimation*.

### G.2  COMPUTATIONAL EFFICIENCY

On a more practical side, although the Beta CDF and its inverse have no closed form solutions for non-integer values of its parameters $\alpha$ and $\beta$, accurate approximations are implemented in many statistical software packages. Then, sampling from Beta distributions with different parameters for each pair of points cannot be done directly on GPU. Coefficients sampled for each pair need to be sent to GPU at each batch, slowing the training because of CPU-GPU synchronizations. An efficient implementation is to define a single `torch.distributions.Beta` with parameters $\alpha = \beta = 1$ (or `Uniform`) on GPU, and then compute a linear approximation of the inverse Beta CDF from *precomputed lookup tables*, which is common when using inverse transform sampling. We present a comparison in Table 8, in terms of performance (accuracy and calibration) and computation time, of the three different variants of implementation discussed when training a ResNet34 on CIFAR10 on a single A100 GPU ($\tau_{max} = 1, \tau_{std} = 0.4$). The three variants achieve comparable performance, and the time difference is in favour of *using warping functions and lookup tables*.

## H  DETAILED EXPERIMENTAL SETTINGS

**Image Classification**  On CIFAR10 and CIFAR100, we use SGD as the optimizer with a momentum of 0.9 and weight decay of $10^{-4}$, a batch size of 128, and the standard augmentations `random crop`, `horizontal flip` and `normalization`. Models are trained for 200 epochs, with an initial learning rate of 0.1 divided by a factor 10 after 80 and 120 epochs. On Tiny-ImageNet, models are trained for 100 epochs using SGD with an initial learning rate of 0.1 divided by a factor of 10 after 40 and 60 epochs, a momentum of 0.9 and weight decay of $10^{-4}$. We use a batch size of 64 and the same standard augmentations. On Imagenet, ResNet-50 (RN50) models are trained for 100 epochs using SGD with an initial learning rate of 0.1, a cosine annealing scheduler, a momentum of 0.9 and weight decay of $10^{-4}$. We use a total batch of 256 split over 3 GPUs and standard augmentations. For ViT models, we train them for 300 epochs using AdamW optimizer, an initial learning rate of 0.01 divided by a factor 10 after 100 and 200 epochs, with a total batch size of 128 split over 3 GPUs.
**Regression**  Following Yao et al. (2022a), we train a three-layer fully connected network augmented with Dropout (Srivastava et al., 2014) on Airfoil, and LST-Attn (Lai et al., 2018) on Exchange-Rate and Electricity. All models are trained for 100 epochs with the Adam optimizer (Kingma & Ba, 2014), with a batch size of 16 and learning rate of 0.01 on Airfoil, and a batch size of 128 and learning rate of 0.001 on Exchange-Rate and Electricity. To estimate variance for calibration, we rely on MC Dropout (Gal & Ghahramani, 2016) with a dropout of 0.2 and 50 samples.
**Method-specific hyperparameters**  On *classification datasets*, we used $\tau_{max} = 1$ and $\tau_{std} = 0.25$ unless stated otherwise. We used the hyperparameters provided by the authors to reproduce state-of-the-art methods, namely, $\alpha = 1$ and $\Delta\lambda > 0.5$ for MIT-A (Wang et al., 2023), and $w = 0.1$, $\alpha = 2.0$ and $\alpha = 1.0$ for CIFAR10/100 and Tiny-ImageNet respectively, and $Q = 4$, for RankMixup (M-NDCG) (Noh et al., 2023). To compare with Mixup, we use $\alpha = 0.2$, the best performing one found in Thulasidasan et al. (2019), and the same value for Manifold Mixup (Verma et al., 2019). On Imagenet, we set $\tau_{max} = 0.1$ and $\tau_{std} = 0.25$, and $\alpha = 0.1$. When using SK Mixup as a regularization term (in SK RegMixup), we set $\tau_{max} = 10$ and keep the same $\tau_{std}$. On *regression datasets*, we used $(\tau_{max}, \tau_{std}) = (0.0001, 0.5)$ on Airfoil, $(\tau_{max}, \tau_{std}) = (5, 1)$ on Exchange Rate, and $(\tau_{max}, \tau_{std}) = (2, 0.2)$ on Electricity. We followed results from Yao et al. (2022a), and fixed $\alpha = 0.5$ on Airfoil, $\alpha = 1.5$ on Exchange Rate and $\alpha = 2$ on Electricity, for Mixup, Manifold Mixup and C-Mixup. Additionally, we searched for the best *bandwidth* parameter for C-Mixup with cross-validation, and found 0.01 on Airfoil, 0.05 on Exchange Rate, and 0.5 on Electricity. Likewise, we searched for the best $\alpha$ for RegMixup, and found $\alpha = 0.5$ on Airfoil, $\alpha = 10$ on Exchange-Rate and $\alpha = 10$ on Electricity.

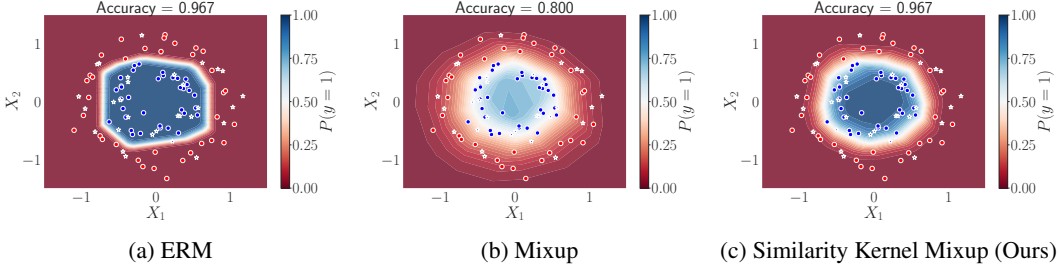

(a) ERM  (b) Mixup  (c) Similarity Kernel Mixup (Ours)

Figure 4: Decision frontiers and data used during training (*circles*) and testing (*stars*) for (a) ERM, (b) Mixup, and (c) our Similarity Kernel Mixup, on Circles toy dataset.

Table 9: Comparison of Performance (Accuracy in %) and calibration (ECE, Brier, NLL) *after Temperature Scaling (TS)* with Resnet34 on CIFAR10 and CIFAR100. Best in **bold**, second best underlined.

| Methods | $\alpha$ | $\tau_{std}$ | CIFAR10 | | | | CIFAR100 | | | |
|---|---|---|---|---|---|---|---|---|---|---|
| | | | Accuracy ($\uparrow$) | TS ECE ($\downarrow$) | TS Brier ($\downarrow$) | TS NLL ($\downarrow$) | Accuracy ($\uparrow$) | TS ECE ($\downarrow$) | TS Brier ($\downarrow$) | TS NLL ($\downarrow$) |
| ERM | – | – | $94.69 \pm 0.27$ | $0.82 \pm 0.11$ | $8.07 \pm 0.31$ | $17.50 \pm 0.61$ | $73.47 \pm 1.59$ | $2.54 \pm 0.15$ | $36.47 \pm 2.05$ | $100.82 \pm 6.93$ |
| | 1 | – | $95.97 \pm 0.27$ | $1.36 \pm 0.13$ | $6.53 \pm 0.36$ | $16.35 \pm 0.72$ | $78.11 \pm 0.57$ | $2.49 \pm 0.19$ | $31.06 \pm 0.69$ | $87.94 \pm 1.98$ |
| Mixup | 0.5 | – | $95.71 \pm 0.26$ | $1.33 \pm 0.08$ | $7.03 \pm 0.46$ | $17.47 \pm 1.18$ | $77.14 \pm 0.67$ | $2.7 \pm 0.36$ | $32.01 \pm 0.93$ | $91.22 \pm 3.05$ |
| | 0.1 | – | $95.37 \pm 0.22$ | $1.13 \pm 0.11$ | $7.37 \pm 0.36$ | $17.43 \pm 0.79$ | $76.01 \pm 0.62$ | $2.54 \pm 0.24$ | $33.41 \pm 0.57$ | $93.96 \pm 1.76$ |
| | 1 | – | $95.16 \pm 0.22$ | $0.6 \pm 0.11$ | $7.3 \pm 0.33$ | $15.56 \pm 0.67$ | $74.44 \pm 0.49$ | $2.02 \pm 0.14$ | $35.25 \pm 0.43$ | $96.5 \pm 1.62$ |
| Mixup IO | 0.5 | – | $95.31 \pm 0.17$ | $0.58 \pm 0.06$ | $7.12 \pm 0.21$ | $\mathbf{15.09 \pm 0.45}$ | $74.45 \pm 0.6$ | $1.94 \pm 0.09$ | $35.2 \pm 0.58$ | $96.75 \pm 1.89$ |
| | 0.1 | – | $95.12 \pm 0.21$ | $0.7 \pm 0.09$ | $7.38 \pm 0.27$ | $15.76 \pm 0.55$ | $74.21 \pm 0.46$ | $2.39 \pm 0.11$ | $35.38 \pm 0.48$ | $98.24 \pm 1.81$ |
| | | 0.2 | $96.29 \pm 0.07$ | $1.35 \pm 0.14$ | $6.37 \pm 0.11$ | $15.97 \pm 0.22$ | $78.13 \pm 0.52$ | $\mathbf{1.31 \pm 0.23}$ | $31.18 \pm 0.72$ | $85.38 \pm 1.92$ |
| SK Mixup | | 0.4 | $\mathbf{96.42 \pm 0.24}$ | $\underline{0.53 \pm 0.06}$ | $\mathbf{6.02 \pm 0.41}$ | $\underline{15.28 \pm 0.92}$ | $78.86 \pm 0.83$ | $\underline{1.42 \pm 0.19}$ | $\underline{30.1 \pm 1.17}$ | $\underline{84.22 \pm 3.35}$ |
| *(Ours)* | 1 | 0.6 | $\underline{96.36 \pm 0.09}$ | $\mathbf{0.52 \pm 0.08}$ | $\underline{6.12 \pm 0.09}$ | $15.7 \pm 0.29$ | $78.63 \pm 0.27$ | $1.74 \pm 0.30$ | $30.41 \pm 0.30$ | $84.97 \pm 1.12$ |
| | | 0.8 | $96.00 \pm 0.41$ | $0.56 \pm 0.05$ | $6.63 \pm 0.58$ | $16.64 \pm 1.27$ | $\mathbf{79.12 \pm 0.52}$ | $1.66 \pm 0.16$ | $\mathbf{29.75 \pm 0.57}$ | $\mathbf{82.82 \pm 1.62}$ |
| | | 1.0 | $96.25 \pm 0.07$ | $0.55 \pm 0.12$ | $6.31 \pm 0.18$ | $16.14 \pm 0.55$ | $78.51 \pm 0.94$ | $1.49 \pm 0.03$ | $30.46 \pm 1.04$ | $85.06 \pm 3.18$ |

# I  ADDITIONAL RESULTS

## I.1  VISUALIZATION

We provide another visualization in Figure 4 on the Circles toy dataset. As interpolations of data from the external circle can fall into the inner one, training with Mixup lead to worst confidence and performance compared to standard ERM, but our SK Mixup recover the accuracy while achieving more meaningful decision boundaries and confidence scores.

## I.2  EFFECT OF $\tau_{STD}$

We present in Table 9 results associated to Table 1 in the main text, but *after temperature scaling*. We have similar observations than discussed in Section 4.2.

## I.3  COMPARISON WITH STATE OF THE ART

We show in Table 10 calibration and predictive performance results associated to Table 2, using the AECE calibration metric, before and after temperature scaling.

Table 10: Comparison of performance (Accuracy in %) and calibration (AECE and AECE after Temperature Scaling), with Resnet101 on CIFAR10, CIFAR100 and Tiny-Imagenet datasets. †: Results reported from Noh et al. (2023). Best in **bold**, second best underlined, *for reported results and ours separately*.

| Methods | CIFAR10 | | | CIFAR100 | | | Tiny-Imagenet | | |
|---|---|---|---|---|---|---|---|---|---|
| | Acc. (↑) | AECE (↓) | TS AECE (↓) | Acc. (↑) | AECE (↓) | TS AECE (↓) | Acc. (↑) | AECE (↓) | TS AECE (↓) |
| MMCE† | 94.99 | 3.88 | 12.9 | **77.82** | 13.42 | 2.8 | **66.44** | 3.38 | 3.38 |
| ECP† | 93.97 | 4.40 | 1.70 | 76.81 | 13.42 | 3.04 | 66.20 | 2.70 | 2.70 |
| LS† | 94.18 | 3.85 | 3.10 | 76.91 | 7.87 | 4.55 | 65.52 | 2.92 | 2.72 |
| FL† | 93.59 | 3.23 | 1.37 | 76.12 | **3.22** | 2.51 | 64.02 | 2.09 | 2.09 |
| FLSD† | 93.26 | 3.67 | **0.94** | 76.61 | 3.29 | **2.04** | 64.02 | 1.81 | 1.81 |
| CRL† | 95.04 | 3.73 | 2.03 | 77.60 | 7.14 | 3.31 | 65.87 | 3.56 | **1.52** |
| CPC† | **95.36** | 4.77 | 2.37 | 77.50 | 13.28 | 3.23 | **66.44** | 3.74 | 3.74 |
| MbLS† | 95.13 | **3.25** | 3.25 | 77.45 | 6.52 | 6.52 | 65.81 | **1.68** | 1.68 |
| ERM | $93.97 \pm 0.03$ | $4.30 \pm 0.01$ | $\mathbf{0.63 \pm 0.01}$ | $74.27 \pm 0.37$ | $14.32 \pm 0.17$ | $1.82 \pm 0.05$ | $65.7 \pm 0.07$ | $4.00 \pm 0.24$ | $1.57 \pm 0.20$ |
| Mixup | $94.78 \pm 0.13$ | $2.73 \pm 0.02$ | $2.59 \pm 0.02$ | $77.50 \pm 0.18$ | $5.77 \pm 2.78$ | $2.67 \pm 0.02$ | $67.80 \pm 0.54$ | $5.45 \pm 1.15$ | $1.68 \pm 0.01$ |
| Manifold Mixup | $94.75 \pm 0.03$ | $2.62 \pm 0.02$ | $2.61 \pm 0.03$ | $77.76 \pm 1.35$ | $5.32 \pm 0.04$ | $3.16 \pm 0.17$ | $68.57 \pm 0.04$ | $6.55 \pm 0.58$ | $2.06 \pm 0.01$ |
| RegMixup | $95.89 \pm 0.01$ | $\mathbf{1.55 \pm 0.02}$ | $1.30 \pm 0.06$ | $78.85 \pm 0.08$ | $6.70 \pm 0.09$ | $2.67 \pm 0.08$ | $69.76 \pm 0.07$ | $10.19 \pm 0.06$ | $1.0 \pm 0.03$ |
| RankMixup | $94.42 \pm 0.17$ | $2.82 \pm 0.22$ | $\mathbf{0.63 \pm 0.14}$ | $77.27 \pm 0.08$ | $11.69 \pm 0.22$ | $2.04 \pm 0.19$ | $65.40 \pm 0.01$ | $13.85 \pm 9.60$ | $1.44 \pm 0.06$ |
| MIT-A | $95.27 \pm 0.01$ | $2.37 \pm 0.02$ | $1.59 \pm 0.01$ | $77.23 \pm 0.56$ | $9.54 \pm 0.48$ | $2.32 \pm 0.05$ | $68.03 \pm 0.09$ | $7.84 \pm 0.10$ | $1.57 \pm 0.05$ |
| SK Mixup *(Ours)* | $95.04 \pm 0.07$ | $2.38 \pm 0.05$ | $0.98 \pm 0.05$ | $78.20 \pm 0.46$ | $\mathbf{2.37 \pm 0.34}$ | $\mathbf{1.38 \pm 0.11}$ | $67.60 \pm 0.01$ | $\mathbf{3.17 \pm 0.22}$ | $1.19 \pm 0.02$ |
| SK RegMixup *(Ours)* | $\mathbf{96.02 \pm 0.04}$ | $2.40 \pm 0.50$ | $1.05 \pm 0.03$ | $\mathbf{79.57 \pm 0.03}$ | $4.34 \pm 1.0$ | $1.66 \pm 0.11$ | $69.11 \pm 0.06$ | $7.17 \pm 0.42$ | $1.06 \pm 0.10$ |

Table 11: Comparison of Accuracy and ECE, before and after Temperature Scaling (TS), with cross-validation when varying $\tau_{\text{std}}$. Results are obtained with a Resnet50 and averaged over 4 splits.

| Dataset | $\tau_{\text{std}}$ | Accuracy | ECE (↓) | TS ECE (↓) |
|---|---|---|---|---|
| CIFAR10 | 0.25 | 95.18 | 2.87 | **0.97** |
| | 0.4 | 95.23 | 3.31 | 1.42 |
| | 0.5 | 95.37 | **2.7** | 2.08 |
| | 0.6 | 95.45 | 3.15 | 1.99 |
| | 0.75 | **95.57** | 3.44 | 2.05 |
| | 0.8 | 95.44 | 3.28 | 2.07 |
| | 1 | **95.57** | 3.44 | 2.05 |
| | 1.25 | 95.46 | 2.71 | 2.25 |
| | 1.5 | 95.29 | 3.42 | 2.36 |
| | 2 | 94.93 | 3.24 | 2.43 |
| CIFAR100 | 0.25 | 77.74 | 3.98 | **1.77** |
| | 0.4 | 78.26 | 4.24 | 2.4 |
| | 0.5 | **78.37** | 3.45 | 3.61 |
| | 0.6 | 78.29 | **2.9** | 4.08 |
| | 0.75 | 78 | 3.65 | 3.91 |
| | 0.8 | 78.03 | 3.33 | 4.27 |
| | 1 | 77.35 | 4.82 | 3.4 |
| | 1.25 | 77.31 | 3.98 | 3.88 |
| | 1.5 | 77.92 | 4.19 | 4.06 |

## J ON THE SELECTION AND SENSITIVITY TO HYPERPARAMETERS

As discussed in Section 3.2, we always draw initial interpolation parameters $\lambda$ from Beta(1,1), removing $\alpha$ as a hyperparameter. Then, $\tau_{max}$ and $\tau_{std}$ can be tuned separately, and, as mentioned in Section 4.2, we found that impact on calibration was mainly controlled by $\tau_{std}$. We selected the values giving the best trade-off between accuracy and calibration using cross-validation, with a stratified sampling on a 90/10 split of the training set, similarly to Pinto et al. (2022), and average the results across 4 different splits. We detail in Table 11 cross-validation results (Accuracy and ECE) showing sensitivity to $\tau_{std}$ on CIFAR10 and CIFAR100 datasets, using a ResNet50. These results lead us to using $\tau_{std} = 0.25$ for all image classification experiments, and show that our approach is not that sensitive to hyperparameter choice between datasets. For Imagenet, we set $\tau_{max} = 0.1$ to follow the commonly used value of $\alpha = 0.1$ in Mixup for this dataset.

Table 12: Performance (RMSE, MAPE) and calibration (UCE, ENCE) comparison for different warping functions on Airfoil dataset. Averaged over 10 random seeds.

| Warping | RMSE ($\downarrow$) | MAPE ($\downarrow$) | UCE ($\downarrow$) | ENCE ($\downarrow$) |
|---|---|---|---|---|
| Sigmoid | $2.892 \pm 0.343$ | $1.733 \pm 0.229$ | $117.95 \pm 45.26$ | $0.018 \pm 0.008$ |
| Beta CDF | $2.807 \pm 0.261$ | $1.694 \pm 0.176$ | $126.02 \pm 23.32$ | $\mathbf{0.018 \pm 0.005}$ |
| Inverse Beta CDF | $\mathbf{2.707 \pm 0.199}$ | $\mathbf{1.609 \pm 0.137}$ | $\mathbf{93.79 \pm 25.46}$ | $0.019 \pm 0.008$ |

Table 13: Performance (RMSE, MAPE) and calibration (UCE, ENCE) comparison for different similarity distance on Airfoil dataset. Averaged over 10 random seeds.

| Warping | RMSE ($\downarrow$) | MAPE ($\downarrow$) | UCE ($\downarrow$) | ENCE ($\downarrow$) |
|---|---|---|---|---|
| Input distance | $2.848 \pm 0.355$ | $1.706 \pm 0.215$ | $105.23 \pm 26.32$ | $0.018 \pm 0.008$ |
| Embedding distance | $2.737 \pm 0.205$ | $1.636 \pm 0.142$ | $101.62 \pm 21.94$ | $\mathbf{0.016 \pm 0.005}$ |
| Label distance | $\mathbf{2.707 \pm 0.199}$ | $\mathbf{1.609 \pm 0.137}$ | $\mathbf{93.79 \pm 25.46}$ | $0.019 \pm 0.008$ |

## K ON THE CHOICE OF SIMILARITY KERNEL

### K.1 ON WARPING FUNCTIONS

As discussed in Section 3.3, the choice of the similarity kernel is highly dependent on the warping function, and more specifically to the correlation between the warping parameter and the shape of the warping function. Given our choice of using the inverse of the Beta CDF as $\omega_\tau$, and that its shape is logarithmically correlated to $\tau$, the choice of a Gaussian kernel seems natural to have an exponential correlation with the distance. Then, the normalization and centering avoid dealing with different range of embedding values between datasets and architectures when choosing a correct value of $\tau_{std}$. Regarding the choice of warping function, as mentioned in Section 3.2, any bijection with a sigmoidal shape could be considered. Besides the inverse of Beta CDF, we also tried the Beta CDF and the Sigmoid ($\lambda \mapsto \frac{1}{1+e^{\lambda/\tau}}$). As they all have logarithmic correlations wrt $\tau$, we used our Gaussian similarity kernel for the three of them. We compare performance on Airfoil dataset after finding the best parameters in each case in Table 12. The inverse of Beta CDF have both the advantage of better results, while preserving the underlying Beta distribution by inverse transform sampling, as we show in Proposition 3.3.

### K.2 ON EMBEDDING SPACES AND DISTANCES

#### K.2.1 REGRESSION TASKS

As discussed in Section 3.3, in regression tasks, we considered either input, embedding or label distance:

- **Input distance:** $\bar{d}_n(\mathbf{x}_i, \mathbf{x}_{\sigma(i)})$,
- **Embedding distance:** $\bar{d}_n(h(\mathbf{x}_i), h(\mathbf{x}_{\sigma(i)}))$,
- **Label distance:** $\bar{d}_n(y_i, y_{\sigma(i)})$.

Input and label distances both have the advantages of inducing almost no computational overhead, as opposed to embedding distance that requires an additional forward pass in the network. We report results comparing the three options on Airfoil dataset in Table 13. We found that label distance was the best performing one, which is in line with results from Yao et al. (2022a), and what we use in the experiments in Section 4.3.

#### K.2.2 CLASSIFICATION TASKS

In classification tasks, as we lack a meaningful distance between label, we restricted our choice between input and embedding distance. Even though computing distances directly in the input

Table 14: Comparison of performance (Accuracy in %), calibration (ECE) after Temperature Scaling, for two different similarity distance. All experiments are using a ResNet34 on CIFAR10 and CIFAR100 datasets and averaged over 4 different random seeds.

| Dataset | Distance | $\tau_{\text{std}}$ | Accuracy ($\uparrow$) | ECE ($\downarrow$) |
|---|---|---|---|---|
| CIFAR10 | Input distance | 0.2 | $95.82 \pm 0.19$ | $\mathbf{0.41 \pm 0.13}$ |
| | | 0.4 | $96.10 \pm 0.17$ | $0.46 \pm 0.06$ |
| | | 0.6 | $95.87 \pm 0.47$ | $0.6 \pm 0.13$ |
| | | 0.8 | $96.12 \pm 0.15$ | $0.70 \pm 0.08$ |
| | | 1.0 | $96.01 \pm 0.27$ | $0.74 \pm 0.19$ |
| | Embedding distance | 0.2 | $96.29 \pm 0.07$ | $1.35 \pm 0.14$ |
| | | 0.4 | $\mathbf{96.42 \pm 0.24}$ | $0.53 \pm 0.06$ |
| | | 0.6 | $96.36 \pm 0.09$ | $0.52 \pm 0.08$ |
| | | 0.8 | $96.0 \pm 0.41$ | $0.56 \pm 0.05$ |
| | | 1.0 | $96.25 \pm 0.07$ | $0.55 \pm 0.12$ |
| CIFAR100 | Input distance | 0.2 | $78.43 \pm 0.52$ | $1.51 \pm 0.10$ |
| | | 0.4 | $78.24 \pm 0.35$ | $1.35 \pm 0.22$ |
| | | 0.6 | $78.36 \pm 0.97$ | $1.5 \pm 0.20$ |
| | | 0.8 | $78.50 \pm 0.41$ | $1.5 \pm 0.16$ |
| | | 1.0 | $78.50 \pm 0.83$ | $1.53 \pm 0.08$ |
| | Embedding distance | 0.2 | $78.13 \pm 0.52$ | $\mathbf{1.31 \pm 0.23}$ |
| | | 0.4 | $78.86 \pm 0.83$ | $1.42 \pm 0.19$ |
| | | 0.6 | $78.63 \pm 0.27$ | $1.74 \pm 0.30$ |
| | | 0.8 | $\mathbf{79.12 \pm 0.52}$ | $1.66 \pm 0.16$ |
| | | 1.0 | $78.51 \pm 0.94$ | $1.49 \pm 0.03$ |

space is faster than relying on embeddings, the latter achieves a better trade-off between accuracy and calibration improvements, which motivated us to use embedding distances as similarity in our experiments in Section 4.2. We compare in Table 14 the two distances, for the different value of $\tau_{std}$, using a Resnet34 on CIFAR10 and CIFAR100 datasets.

Table 15: Comparison of performance (Accuracy in %) and calibration (ECE and ECE after Temperature Scaling), using a simple baseline with Resnet101 on CIFAR10, CIFAR100 and Tiny-Imagenet datasets.

| $\alpha_{\text{low}}$ | $\alpha_{\text{high}}$ | CIFAR10 | | | CIFAR100 | | | Tiny-Imagenet | | |
|---|---|---|---|---|---|---|---|---|---|---|
| | | Acc. ($\uparrow$) | ECE ($\downarrow$) | TS ECE ($\downarrow$) | Acc. ($\uparrow$) | ECE ($\downarrow$) | TS ECE ($\downarrow$) | Acc. ($\uparrow$) | ECE ($\downarrow$) | TS ECE ($\downarrow$) |
| 5 | 0.1 | 95.55 | 1.32 | 0.69 | 79.35 | 5.13 | 1.55 | 66.46 | 1.85 | 1.52 |
| 5 | 0.5 | 95.40 | 6.68 | 0.56 | 76.95 | 6.18 | 1.51 | 67.15 | 7.15 | 1.37 |
| 10 | 0.1 | 93.43 | 2.92 | 1.12 | 75.90 | 1.15 | 1.12 | 67.65 | 1.59 | 1.18 |
| 10 | 0.5 | 95.15 | 5.71 | 0.59 | 76.63 | 6.48 | 1.65 | 67.83 | 6.49 | 1.16 |

Table 16: Comparison of performance (Accuracy in %) and calibration (ECE and ECE after Temperature Scaling), using non-linear mixing with Resnet101 on CIFAR10, CIFAR100 and Tiny-Imagenet datasets.

| Methods | CIFAR10 | | | CIFAR100 | | | Tiny-Imagenet | | |
|---|---|---|---|---|---|---|---|---|---|
| | Acc. ($\uparrow$) | ECE ($\downarrow$) | TS ECE ($\downarrow$) | Acc. ($\uparrow$) | ECE ($\downarrow$) | TS ECE ($\downarrow$) | Acc. ($\uparrow$) | ECE ($\downarrow$) | TS ECE ($\downarrow$) |
| CutMix | 95.89 | 2.14 | 0.84 | 78.58 | 9.49 | 2.99 | 67.79 | 6.25 | 1.62 |
| SK CutMix | 95.25 | 1.96 | 0.5 | 78.65 | 6.70 | 2.21 | 67.86 | 2.94 | 1.98 |

## L    COMPARISON WITH A SIMPLER BASELINE

By defining fixed Beta distributions depending on the distance, we can have a more efficient (in terms of computation time) implementation. However, we also expect worse results as it will lack granularity in the behavior wrt to the distance. We experimented with defining two separate Beta distributions, one with parameter $\alpha_{\text{low}}$, for pairs with distance lower than average, and the other with parameter $\alpha_{\text{high}}$, for pairs with distance higher than average. We present in Table 15, results with the same settings as Table 2. We can see that using lower $\alpha_{\text{high}}$ improves calibration, which confirms the high level idea of reducing interpolation for high-distance pairs. However, finding a good pair of parameters $(\alpha_{\text{low}}, \alpha_{\text{high}})$ is more dataset-dependent than relying on our similarity kernel approach.

## M    COMBINATION WITH NON-LINEAR MIXING

Combining our method (or other calibration-driven method) with non-linear mixup for images, like CutMix (Yun et al., 2019), can be done, since the changes are in different parts to each other. However, extensive experiments in Pinto et al. (2022) (in their Appendix H) have shown that combining their RegMixup with CutMix is not trivial as it does not always improve predictive performance and robustness. We also expect that introducing a non-linear mixing operation will not always lead to the same improvements of calibration with our method, since theoretical groundings are always based on the linear mixing operation. Recent results from Oh & Yun (2023) has also shown that the distortion of the training loss induced by non-linear mixup methods, like CutMix, leads to different and less optimal decision boundaries than the loss obtained with linear mixup. We leave for future work the theoretical derivations of using the non-linear mixing operation with the aim to improve calibration and robustness. Nonetheless, we present in Table 16 first results with CutMix and SK CutMix, in which we apply our similarity kernel to change dynamically the Beta distribution in CutMix, with the same settings as Table 2. We can see that SK CutMix can improve accuracy over CutMix. Although SK CutMix can also improve calibration over CutMix, since CutMix does not improve calibration over Mixup, SK CutMix does not lead to better calibration than SK Mixup.

Table 17: Efficiency comparison between Mixup methods. We report the number of batch of data in memory at each iteration, the best epoch measured on validation data, time per epoch (in seconds) and total training time to reach the best epoch (in seconds).

(a) Image classification on CIFAR10, CIFAR100 and Tiny-imagenet datasets, with a Resnet50.

| Method | Batch in memory | CIFAR10 | | | CIFAR100 | | | Tiny-Imagenet | | |
|---|---|---|---|---|---|---|---|---|---|---|
| | | Best Epoch | Time | Total Time | Best Epoch | Time | Total Time | Best Epoch | Time | Total Time |
| Mixup | 1 | 186 | 17 | 3162 | 181 | 17 | 3077 | 89 | 125 | 11125 |
| CutMix | 1 | 176 | 24 | 4224 | 148 | 24 | 3552 | 69 | 143 | 9867 |
| RankMixup | 4 | 147 | 78 | 11485 | 177 | 78 | 13806 | 80 | 540 | 43380 |
| MIT-A | 2 | 189 | 46 | 8694 | 167 | 46 | 7659 | 82 | 305 | 24908 |
| RegMixup | 2 | 173 | 32 | 5536 | 177 | 32 | 5664 | 94 | 240 | 22560 |
| SK Mixup | 1 | 183 | 28 | 5138 | 160 | 28 | 4480 | 69 | 189 | 13104 |
| SK CutMix | 1 | 196 | 30 | 5880 | 124 | 30 | 3720 | 71 | 185 | 13135 |
| SK RegMixup | 2 | 175 | 39 | 6825 | 179 | 39 | 6981 | 75 | 316 | 23700 |

(b) Time series regression on Electricity dataset.

| Methods | Batch in memory | Best Epoch | Time | Total Time |
|---|---|---|---|---|
| C-Mixup | 2 | 87 | 46.51 | 4046 |
| RegMixup | 2 | 85 | 11.46 | 975 |
| SK Mixup *(Ours)* | 1 | 78 | 11.16 | 867 |

# N FULL COMPUTATIONAL EFFICIENCY COMPARISON

We present in Table 17 the full comparison of efficiency with all baselines. As can be seen, our SK Mixup is about $1.5\times$ *faster* than MIT-A, about $3\times$ faster than RankMixup, and about $4\times$ faster than C-Mixup. Additionally, we use *a single batch of data* for each iteration, while MIT-A requires training on *twice* the amount of data per batch, and *four times* the amount of data per batch for RankMixup. This adds significant memory constraints, which limits the maximum batch size possible in practice. Unlike C-Mixup, our approach does not rely on sampling rates calculated before training, which add a lot of computational overhead and are difficult to obtain for large datasets.

