# OpenReview forum: "Tailoring Mixup to Data for Calibration"
_ICLR.cc/2025/Conference — ICLR 2025 Poster_

### Official Review · Reviewer_ymX4 · 2024-10-27

**Soundness:** 2
**Presentation:** 3
**Contribution:** 2
**Rating:** 6
**Confidence:** 3

**Summary:**

This paper proposes to improve different mixup methods by adjusting the probability distribution to sample interpolation weights according to the distance between a given sample pair. Using the proposed method, sample pairs that are far from each other are less likely to be mixed. Empirical results on different tasks and data sets demonstrate the effectiveness of the proposed method.

**Strengths:**

The proposed method is clearly introduced and easy to understand

**Weaknesses:**

- The motivation of proposed method can be strengthened
- Some baseline methods seem missing, which can make the empirical comparison not supportive enough

**Questions:**

- Instead of using a similarity kernel introduced in this paper, a simple implementation is to divide sample pairs based on their distances (like the case in Figure 2) and use two different Beta distributions to sample the interpolation weights. How will this baseline perform compared to the proposed method? Such comparison can further verify the motivation and support the proposed method.
- While the authors have mentioned that non-linear interpolation methods have several disadvantages (larger computational cost, limited application) in section 2.1, some empirical comparison on their performance for image data sets in Table 2-4 should still be necessary. In addition, the authors may also include some of these interpolation methods in Table 6 to verify that their computational cost are muchh larger compared to linear mixup methods.
- Moreover, can the proposed method be combined with these non-linear interpolation methods as well? I suppose such combination will be straight-forward, as we only need to change the sampling distribution of interpolation weights for different sample pairs. Some discussion (possibly with some empirical results) will be welcome here.
- I am a bit puzzled by the performance of different methods on ImageNet-A in Table 4. None of them has an accuracy higher than 3%, which seems worse than most methods reported in [1]. Is this due to some discrepancies in experimental setup (e.g., less training epochs)? If so, the authors may need to clarify that such discrepancies are not in favor of their proposed method.
- Also, how are the baseline methods chosen in Table 6? From my perspective, I suppose we need to compare the time cost of SK Mixup against Mixup or some other baseline methods with rather good performance. Nevertheless, it seems either of the above cases applies, and some explanation may be needed here.

## References
[1] On Feature Normalization and Data Augmentation. CVPR 2021

---

> ### Author Response · Authors · 2024-11-25
> **Rebuttal 1/2**
>
> We thank the reviewer for their review. We answer to their questions and remarks below.
>
> - **W1**: In the revision, we introduce a new corollary derived from Theorem 5.1 in Liu et al. [A], showing that the probability of assigning a wrong label with mixup is lower bounded by the distance between the manifolds of the original data used for mixing. We refer the reviewer to our answer **W1** to reviewer XhfP for the full derivation and discussion and Theorem 3.2 in the revision. This novel theoretical result, which is confirmed by the experiments in Figure 1a, better supports and motivates our method.
> - **W2**: We compare our method to recently published Mixup methods designed to improve calibration on a wide range of benchmarks.
> - **Q1**: This is a good suggestion. By defining fixed Beta distributions depending on the distance, we can have a more efficient (in terms of computation time) implementation. However, we also expect worse results as the method lacks granularity in its behavior.
>   We present below experiments with a ResNet-101 on CIFAR 10 and CIFAR 100, with the same settings as results in Table 2. We defined two separate Beta distributions, one with $\alpha_{low}$ parameter, for pairs with distance lower than average, and the other with $\alpha_{high}$ parameter, for pairs with distance higher than average.
>   We will add this comparison and discussion in the Appendix.
>
> Dataset | $\alpha_{low}$ | $\alpha_{high}$ | Acc. | ECE | TS ECE
> --- | --- | --- | --- | --- | ---
> CIFAR 10 | 10 | 0.1 | 93.43 | 2.92 | 1.12
> |  | 5 | 0.5 | 95.40 | 6.68 | 0.56
> CIFAR 100 | 10 | 0.1 | 75.90 | 1.15 | 1.12
> |  | 5 | 0.5 | 76.95 | 6.18 | 1.51
>
> - **Q2 + Q3**: Combining our method (or other calibration-driven method) with non-linear mixup for images, like CutMix [B], can be done, since the changes are in different parts to each other. However, extensive experiments in Pinto et al. [C] (in their Appendix H) have shown that combining their method with CutMix is not trivial as it does not always improve results and robustness. We also expect that introducing a non-linear mixing operation will not lead to the same improvements of calibration with our method, since theoretical groundings and intuitions are always based on the linear mixing operation. Recent results [D] has also shown that the distortion of the training loss induced by non-linear mixup methods, like CutMix, leads to different and less optimal decision boundaries. We leave for future work the theoretical derivations of using the non-linear mixing operation with the aim to improve calibration and robustness.
>   Nonetheless, we present below first results with CutMix and SK CutMix, in which we apply our similarity kernel to change dynamically the Beta distribution in CutMix, in the same settings as Table 2. We can see that SK CutMix can improve accuracy over CutMix. Although SK CutMix can also improve calibration over CutMix, since CutMix does not improve calibration over Mixup, SK CutMix does not lead to better calibration than SK Mixup. We will add this discussion and a full comparison in the revision.
>
> Dataset | Methods | Acc. | ECE | TS ECE
> --- | --- | --- | --- | ---
> CIFAR 10 | CutMix | 95.89 | 2.14 | 0.84
> | | SK CutMix | 95.25 | 1.96 | 0.5
> CIFAR 100 | CutMix | 78.58 | 9.49 | 2.99
> | | SK CutMix | 78.65 | 6.70 | 2.21
> Tiny ImageNet | CutMix | 67.79 | 6.25 | 1.62
> | | SK CutMix | 67.86 | 2.94 | 1.98
>
> - **Q4**: In the reference, methods are evaluated either with pretrained checkpoints from internet, or with ResNet-50 trained with 300 epochs. In our experiments on ImageNet, we trained all methods for 100 epochs on Imagenet, following similar settings as Pinto et al. [C] (detailed training settings in our Appendix H).
>   We also trained a ViT for 300 epochs on Imagenet (detailed training settings in our Appendix H), and we obtain better results on ImageNet A. We show the complete benchmark with ViT in the table below.
>
>   Methods | IN Acc | IN ECE | IN AECE | IN-R Acc | IN-R ECE | IN-R AECE | IN-A Acc | IN-A ECE | IN-A AECE
>    --- | --- | --- | ---- | --- | --- | ---- | --- | --- | ----
>   ERM | 69.34 | 9.03 | 9.03 | 15.46 | 35.93 | 35.93 | 1.83 | 45.03 | 45.03
>   Mixup | 72.0 | 5.81 | 5.8 | 18.21 | 29.29 | 29.29 | 2.47 | 41.07 | 41.07
>   Manifold Mixup | 72.04 | 6.99 | 6.95 | 18.93 | 30.27 | 30.27 | 2.15 | 43.47 | 43.47
>   RegMixup | _74.44_ | 7.36 | 7.34 | _21.01_ | 32.64 | 32.64 | _3.8_ | 44.4 | 44.4
>   RankMixup | 69.99 | 9.3 | 9.3 | 15.77 | 37.37 | 37.37 | 1.77 | 46.82 | 46.82
>   MIT-A | 72.81 | 5.69 | 5.69 | 17.6 | 31.07 | 31.07 | 2.81 | 41.8 | 41.8
>   SK-Mixup (Ours) | 71.83 | _3.89_ | _3.9_ | 18.37 | _28.56_ | _28.56_ | 2.28 | _39.33_ | _39.33_
>   SK RegMixup (Ours) | **75.11** | **2.0** | **1.98** | **22.06** | **26.23** | **26.23** | **4.44** | **38.25** | **38.25**

---

> > ### Author Response · Authors · 2024-11-25
> > **Rebuttal 2/2**
> >
> > - **Q5 + Q2**: We added the time comparison of all baselines, and non-linear methods, in the table below. We will add this full comparison in the revision.
> >
> > Method | Batch in memory | C10 - Best epoch | C10 - Time / epoch (s) | C10 - Total Time (s) | C100 - Best epoch | C100 - Time / epoch (s) | C100 - Total Time (s) | Tiny-IN - Best epoch | Tiny-IN - Time / epoch (s) | Tiny-IN - Total Time (s)
> > --- | --- | --- | --- | --- | --- | --- | --- | --- | --- | ---
> > Mixup | 1 | 186 | 17 | 3162 | 181 | 17 | 3077 | 89 | 125 | 11125
> > CutMix | 1 | 176 | 24 | 4224 | 148 | 24 | 3552 | 69 | 143 | 9867
> > RankMixup | 4 | 147 | 78 | 11485 | 177 | 78 | 13806 | 80 | 540 | 43380
> > MIT-A | 2 | 189 | 46 | 8694 | 167 | 46 | 7659 | 82 | 305 | 24908
> > RegMixup | 2 | 173 | 32 | 5536 | 177 | 32 | 5664 | 94 | 240 | 22560
> > SK Mixup | 1 | 183 | 28 | 5138 | 160 | 28 | 4480 | 69 | 189 | 13104
> > SK CutMix | 1 | 196 | 30 | 5880 | 124 | 30 | 3720 | 71 | 185 | 13135
> > SK RegMixup | 2 | 175 | 39 | 6825 | 179 | 39 | 6981 | 75 | 316 | 23700
> >
> > We hope that our answer and additional results clarifies your concerns, and would be happy to engage in discussion for any other remaining concern.
> >
> > [A] Liu et al. "Over-training with mixup may hurt generalization." ICLR 2023.
> >
> > [B] Yun et al. Cutmix: Regularization strategy to train strong classifiers with localizable features. In ICCV 2019.
> >
> > [C] Pinto et al. Regmixup: Mixup as a regularizer can surprisingly improve accuracy and out distibution robustness. In NeurIPS 2022.
> >
> > [D] Junsoo Oh and Chulhee Yun. Provable benefit of mixup for finding optimal decision boundaries. In ICML 2023.

---

> > > ### Author Response · Authors · 2024-12-01
> > >
> > > Dear reviewer ymX4, we would be happy to receive your feedback on our rebuttal and the updates on our manuscript before the end of the discussion period.

---

> > > > ### Comment · Reviewer_ymX4 · 2024-12-03
> > > > **Thank you for your responses**
> > > >
> > > > I would like to thank the authors for their detailed responses that have clarified most of my previous concerns. It seems that the authors have incorporated some (but not all) of these additional experiments in their paper, and I would suggest the authors make a thorough revision for the final version of their submission, as several other reviewers share some concerns on the clarity of this work. I have increased my score above the acceptance bar.

---

> > > > > ### Author Response · Authors · 2024-12-03
> > > > >
> > > > > Thank you for the positive update, we appreciate the comments and will take them into account ! We are still running more experiments to have a thorough and complete benchmark with the baseline proposed by the reviewer, which is why we did not incorporate it yet. We will include them as well as the discussion into the final version.

---

### Official Review · Reviewer_Fzdf · 2024-10-31

**Soundness:** 2
**Presentation:** 1
**Contribution:** 2
**Rating:** 6
**Confidence:** 3

**Summary:**

The paper has come up with a concept called manifold mismatch, which is a phenomenon in Mixup that can harm the generalization and calibration performance of the trained models. The paper also show that such a manifold mismatch behavior is correlated to the distance between the data points being paired and mixed in Mixup. Then, by applying a similarity kernel, the paper proposes a variant training algorithm of Mixup that can dynamically adjust the mixing coefficient depending on the distance of the data points being mixed. Finally, the authors have empirically verified the effectiveness of the proposed algorithm on various models and tasks.

**Strengths:**

1. It's a novel idea to summarize several observed phenomenons or properties of Mixup into one unified terminology: manifold mismatch
2. The correlationship between data points pair distance and the occurence of manifold mismatch is carefully derived.
3. The proposed algorithm is meant to balance the diversity and the uncertainty of the virtual examples in Mixup, which is meaningful and helpful

**Weaknesses:**

1. Poor presentation
2. Doesn't clarify the connection between manifold mismatch and calibration.
3. The core claim that distance impacts manifold mismatch is too simple to be seen, and it's actually under-justified since there seems to be much more necessary conditions of manifold mismatch like data distribution, structure of learned features and manifolds, etc.. In fact, the idea of improving Mixup by forcing more mixtures between closer points is not novel, like k-mixup [1]
4. The idea of dynamically adjust the mixing coeffecient is also not new, like AdaMixup [2].

[1] https://arxiv.org/abs/2106.02933

[2] https://arxiv.org/abs/1809.02499

**Questions:**

1. Line 011 or 012 on page 1. "Along with improved performance", what performance, please make it specified
2. Line 012 or 013 on page 1. "improving calibration and predictive confidence", from my understanding they are the same thing, no need to use "and" here in my opinion.
3. Line 014. Infact in this line the paper has mentioned "calibration of confidence", what is calibration of "confidence", again, in my opinion it's just calibration.
4. Line 019 to 020. Again, please specify what "performance" it is when mentioning "improve performance".
5. Line 054. Here the paper mentions "label noise", can the authors provide a brief description of it like the way they have described manifold intrusion in previous text?
6. Line 140. Again, please specify the term "performance".
7. Is there and related research works about calibration-driven Mixup methods in regression tasks?
8. Line 161. If possible it's much better to turn this title to the next page, it would look better.
9. Line 162 to 169. The notation preliminaries in this paragraph feel a bit too simple and careless. For example, when defining a dataset, one may want to first define a data space and label space, indicating that these spaces are subspaces of some vector spaces of certain dimensions, and then indicate that the training dataset is of some size with the examples being drawn from some data distribution over the data space, etcs.. Putting those details in one single line feels careless. Also, what is $M$? Many would of course just simply take it as the number of classes in the classification problems, but it also needs to be specified in the paper.
10. Line 162 to 169. There is also some notations that are inconsistent. The paper first indicates that the labels are $M$-dim vectors, then what is the input and output dimension of the encoder $h_\phi$? The way it is presented make the model output $f_\theta(\text{x})$ feels like a scalar, while it should be a vector the same dimension as the labels since $\hat{y}:=f_\theta(\text{x})$.
11. Line 193 to 194. "bounded support $\mathcal{M}_m\subset\mathcal{H}$". Is this also how the manifolds are defined in this paper?
12. Line 194 to 195. To assume that classes are separated, it means all samples belonging to "different classes manifolds" should disjoint, is that correct?
13. Line 208. Why would the mixed points' belonging "to no vlass manifold at all" be a bad thing? In my opinion, when using Mixup for training, especially when the data dimension is high, it's actually mostly the case that the mixed point fall into the "void" areas in the entire data space.
14. Line 212. "the higher the distance between two points, the more likely their convex combination will fall outside of the original manifolds". Here is a counter example. Suppose we have two classes of points in $\mathbb{R}^2$. Suppose their manifolds are two separated line segments on the horizontal axis. If we combine the rightest point from the left manifold and the leftest point from the right manifold, the mixed points will almost all fall into the middle area which is outside both the original manifolds. If we combine the leftest points from both manifold, the fact is, a much bigger proportion of mixed points would fall inside the left manifold. But, the distance between the points in the second case is much larger than that in the first case.
15. Line 215. "Using a Resnet18 trained ...", is it trained using ERM or Mixup?
16. How to choose $\tau_{std}$ and $\tau_{max}$?
17. How does SK Mixup help improve calibration? The derivation of the algorithm only suggests that SK Mixup can help mitigate manifold mismatch, but is manifold mismatch the necessary reason, or even the real reason, of bad calibration?
18. In the process of trading off between diversity and uncertainty, how would calibration and generalization behave? Will they behave like a up-side-down U curve such that there is a sweet spot?
19. The page number of page 7, and the header "under review" line in page 8, there are some strange hyperlinks around the text.
20. Theorem 3.1 (i). "$\lambda\in]\lambda_1,\lambda_2[$", it should be $[\lambda_1,\lambda_2]$ right?

---

> ### Author Response · Authors · 2024-11-25
> **Rebuttal 1/2**
>
> We thank the reviewer for the review and the detailed comments. We answer point by point below.
>
> - **W1**: We took into account all remarks and improved the presentation of the paper.
> - **W2**: We clarify the connections, from the discussions and results in the paper:
>   - We define manifold mismatch as events where mixed samples are outside their original class manifold. Being outside the class manifold can lead to label noise or manifold intrusion. Thus, events of "label noise for a mixed sample" are included in events of "manifold mismatch for a mixed sample".
>   - Assignment to a wrong label with mixup is linked to distance between points (Figure 1a in the revision), the higher the distance between points, the higher the chance of the mixed sample being associated to a wrong label. These results are also supported by theoretical grounding, that we introduce as a corollary of Theorem 5.1 in Liu et al. [A], which proves that the probability of assigning a noisy label with mixup is lower bounded by the distance between the two manifold. We refer the reviewer to our answer to reviewer XhfP for the full derivation and discussion (Theorem 3.2 in the revision).
>   - Mixing points with lower distance is linked to better calibration (figure 1b), we explain this as mixing only points that are similar reduces occurrence of assigning noisy labels to mixed samples. This improves the quality of labels seen by the model, and its confidence.
>   - We show that SK Mixup reduces likelihood of association to a wrong manifold even for points far away (Figure 2.b), compared to original Mixup.
>   - We show throughout the paper with many experiments on multiple datasets that SK Mixup improves calibration.
>
>   We hope this summary clarifies the connections between manifold mismatch, association to wrong label, distance between points and calibration.
>
> - **W3**: We can directly see in Figure 1a that, empirically, the distance between the two points will impact negatively the likelihood of assigning the correct synthetic label to a mixed sample. We also introduce an additional theoretical result supporting this (Theorem 3.2 in the revision).
>   While the k-mixup method does take into account the distance between two batch of data, it is to *select the points* to mix with, which is not the same as "forcing more mixtures between closer points". This method would be more related to C-Mixup. As we discuss in the paper (related work section and in the end of Section 3.1), restricting pair to mix by selecting data based on distance can degrade predictive performance (accuracy, MSE, etc ...) by reducing the diversity of the possible mixing directions and thus the diversity of synthetic samples. We will include this discussion of this reference in the related work section.
>
> - **W4**: We discuss relation with this work in the related work section (section 2.1). We will reiterate and deepen the discussion. The goal of AdaMixup is to learn the distribution from which mixing coefficient are sampled, and the method is doing so with two external models that predicts the mixing policy to apply depending on each pair of samples. However, as shown in the original paper, the mixing policy is very stable and narrow around 0.5, which means that the predicted mixing coefficients are always very close to 0.5, and that restricts the diversity of synthetic samples. The method is thus fundamentally different from ours, since we explicitly define the mixing distribution as a direct function of the distance.
>
> - **Q1-Q4 + Q6**: We usually referred to "performance" as a general term for "predictive performance" like accuracy in classification, RMSE or MAPE in regression. We will carefully check the paper to make that clearer.
>
> - **Q5**: Label noise can be briefly describes as occurring for a mixed sample when its ground-truth assignment and its mixed label does not match, i.e. when the mixed sample should be associated to a different class than the original ones. A more formal definition of label noise can be found in Liu et al. [A].
>
> - **Q7**: We did not find any reference combining calibration in regression tasks and Mixup.
>
> - **Q8**: Thank you, we fixed that.
>
> - **Q9 + Q10**: We wanted to be concise since these are typical notations and settings used in the literature.
>   We explicitly say that M is the number of classes in line 192, but we will clarify the definition of the notations with the two following small modifications:
>   - for classifications tasks, $\mathbb{Y} = \{ 1,\dots, M \}$, where $M$ is the number of classes,
>   - specify $h_{\varphi}: \mathbb{X} \rightarrow \mathbb{R}^{d_2}$ and $\mathbf{w} \in \mathbb{R}^{d_2 \times M}$, with $d_2$ the embedding dimension, such that $\hat{y} := \arg \max \text{softmax}(f_\theta(x))$, and $f_\theta(x) = \mathbf{w}^T h_{\varphi}(x) \in \{1,\dots, M\}$.
>
>   These do not change the remaining notations in the paper.
>
> - **Q11**: Yes.

---

> > ### Author Response · Authors · 2024-11-25
> > **Rebuttal 2/2**
> >
> > - **Q12**: Yes, we mention it in line 194, the class manifolds are disjoints.
> >
> > - **Q13**: As we discuss in the introduction, a mixed sample being outside the original class manifolds can lead to label noise if it is closer to a different manifold, since the assigned mixed label will not include this different manifold.
> >   The general idea is that events of "label noise for a mixed sample" are included in events of "manifold mismatch for a mixed sample".
> >
> > - **Q14**: We will correct this part into "the higher the distance between the two points, **the less likely their convex combination will fall near the original manifolds**". This is the behavior that we observe empirically in both CIFAR10 and CIFAR100 datasets in Figure 1a, and that we assume is present in practice in other datasets as well.
> >   The high level intuition is that the distance between two points from two different manifolds roughly informs about the average distance between the two manifolds.
> >   Then, the line segment between these two points is more likely to have regions where a different manifold is closer or intersects (label noise or manifold intrusion) if the two original manifolds are far away.
> >   We also derive a theoretical result (theorem 3.2), supporting these observations.
> >
> >   The toy problem discussed is thus limited by-design and cannot illustrate the behavior we are interested in, since there are only two manifolds of data.
> >   Let's introduce a third manifold of data in the example, such that there exists a pair of manifolds whose averaged pairwise distance is higher than those of all the other pairs of manifolds (which avoids unlikely cases of equidistant manifolds). Then, a pair of points having a high distance is more likely to be respectively from these two manifolds, and there will be regions along their line segment that will be closer to the third one.
> >
> > - **Q15**: The ResNet18 was trained using ERM.
> > - **Q16**: We discuss the selection of $\tau_{std}$ and $\tau_{max}$ in Appendix J. Since we found that calibration was mainly impacted by $\tau_{std}$, we fixed $\tau_{max}=1$ and cross validated different values of $\tau_{std}$ on CIFAR10 and CIFAR100 datasets. We found that $\tau_{std} = 0.25$ achieved a good trade-off between accuracy and calibration, and fixed this value for all other experiments in the different datasets of image classification. For Imagenet, we set $\tau_{max} = 0.1$ to follow the commonly used value of $\alpha = 0.1$ in Mixup for this dataset, but kept the same $\tau_{std} = 0.25$ as other datasets.
> > - **Q17**: SK Mixup takes into account the distance between the two points to influence the interpolation strength, which avoids creating synthetic samples far away from either of the original manifolds. Thus, SK Mixup reduces the likelihood of creating synthetic data with a combination of labels that conflicts with distances to manifolds, i.e. synthetic data with a noisy assigned label. We illustrate that in Figure 3.b, where we plot the likelihood of a pretrained model assigning either of the original label to synthetic data created with SK mixup, with respect to the distance between the two points. As can be seen in the plot, the likelihood remains stable with the distance, as opposed to the original behavior observed with Mixup in Figure 1. This shows that, even if the two points are far away, the synthetic label assigned should not contradict with the distance to the other manifolds. By reducing conflicts in synthetic labels, we expect the model to have better predictive confidence and better calibration. This behavior is confirmed in practice in our experiments.
> >
> > - **Q18**: Our results in Table 7, which shows detailed values of results illustrated in Figure 2, using multiple quantiles, would indicate that, with Mixup, increasing diversity improves accuracy while increasing uncertainty decreases calibration. Our goal is thus to improve diversity without adding uncertainty, and our proposed SK Mixup is a step in that direction.
> >
> > - **Q19**: Thank you, this is fixed.
> > - **Q20**: In our definition, $\tilde{x}(\lambda_1)$ and $\tilde{x}(\lambda_2)$ are the respective borders of the two manifold on the line segment. These borders can then be within the manifolds or not, depending on if they are closed or open spaces. Although we assumed they are closed, considering $\lambda \in ]\lambda_2, \lambda_1[$ makes the result true in both case and more general. Including the edge cases of $\tilde{x}(\lambda_1)$ and $\tilde{x}(\lambda_2)$ are not really important to the result.
> >
> > We hope that our answer clarifies your concerns, and would be happy to engage in discussion for any other remaining concern.
> >
> > [A] Liu et al. "Over-training with mixup may hurt generalization." ICLR 2023

---

> > > ### Author Response · Authors · 2024-12-01
> > >
> > > Dear reviewer Fzdf, we would be happy to receive your feedback on our rebuttal and the updates on our manuscript before the end of the discussion period.

---

> > > > ### Comment · Reviewer_Fzdf · 2024-12-01
> > > >
> > > > Thank you for the detailed responses.
> > > >
> > > > 1. I appreciated the changes in the presentations of the paper.
> > > > 2. W2. The discussion of the connection between the distances of the example pairs and the occurence of manifold mismatch still seems to be mostly based on intuitions. In my opinion, this claim itself, is intuitively obvious. One of the other reasons that I mentioned k-mixup is that I am trying to say that this claim is not novel, in fact, it's easy to be noticed. Therefore, if one wants to take this claim as one of the main contributions of their paper, I would prefer to see some more strict justifications. For example, one may mathematically quantify the manifold mismatch behavior and show how the distances of the example pairs may impact this quantity. Without this kind of justifications, I don't think this claim underqualified to be one of the major contributions of an ICLR paper.
> > > > 3. Q14. First, the toy example with 3 classes still doesn't fit the claim. Suppose the three classes are three separated line segments on the horizontal axis. If we take: (a) the righest point of class 1 and leftest point of class 2, and (b) the leftest point of class 1 and rightest point of class 3. In this context, case (a) still have much higher chance of having mixed point near original manifolds than case (b). Second, I personally believe that it's never "limited by-design" to consider some unideal or even worst-case examples of data distribution in algorithm design in general. Unless of course if this paper mainly focuses on large-scaled in-real-life datasets like Cifar10 or Cifar100. But that would require some case-dependent analysis of the data distribution sepecifically made to be applied to these two datasets, which is not seen in this paper. In fact, the data analysis part of this paper is written in a way to appear to be applied to the general cases.
> > > >
> > > > In conclusion, I think this paper is underqualified to be accepted by ICLR. Nevertheless, the idea of balancing the quality and the diversity of the mixed data is meaningful. I would still keep my original rating.

---

> > > > > ### Comment · Reviewer_Fzdf · 2024-12-01
> > > > >
> > > > > Also, about Q20, I was trying to say that the brackest of an interval should be "[ ]" instead of "] [". At least I have never seen the second case before. I was wondering if this is a typo or this has its own meaning? Thank you!

---

> > > > > > ### Author Response · Authors · 2024-12-02
> > > > > >
> > > > > > > Also, about Q20, I was trying to say that the brackest of an interval should be "[ ]" instead of "] [". At least I have never seen the second case before.
> > > > > >
> > > > > > It is not a typo, the notations "$]a,b[$" indicates that the endpoints $a$ and $b$ are excluded from the set. Another equivalent notation would be "$(a,b)$". We refer the reviewer to https://en.wikipedia.org/wiki/Interval_(mathematics)#Notations_for_intervals for a more detailed discussion with references. As we said in our previous answer, including the borders of the manifold does not impact the results and is more general as it will otherwise depend on the manifolds being considered as closed or open sets.

---

> > > > > > > ### Comment · Reviewer_Fzdf · 2024-12-02
> > > > > > >
> > > > > > > Thank you for the detailed explanation. I would appreciate if the authors can add a condition "$M>2$" in the paper and also an explanation of the "]a,b[" notation in the preliminary part.
> > > > > > >
> > > > > > > For now, I think the updated version of the paper is indeed well written, and the claims are sufficiently justified, especially considering the comprehensiveness of the experiments part, I have raised my rating to 6, but lowered the confidence to 3.

---

> > > > > > > > ### Author Response · Authors · 2024-12-03
> > > > > > > >
> > > > > > > > Thank you very much for the positive update and comments ! We will add both in the preliminary part. We were also not aware that there were other notations in use for "excluding endpoints of intervals".

---

> ### Author Response · Authors · 2024-12-02
>
> We thank the reviewer for the quick reply. We are glad to know that the reviewer appreciates our changes on the presentation and that our idea is meaningful. However, it is not clear to us what more justifications the reviewer is expecting.
>
> > The discussion of the connection between the distances of the example pairs and the occurence of manifold mismatch still seems to be mostly based on intuitions.
>
> The discussion is not "mostly based on intuition", **we have two theoretical results supporting it**. We prove: (i) the existence of manifold mismatch for any pair of manifold (theorem 3.1) and (ii) the connections between wrong label association (which is linked to manifold mismatch) and distance between manifolds (theorem 3.2). **We also present a wide range of experiments showing that taking distance into account improves calibration from state of the art**.
>
> > In my opinion, this claim itself, is intuitively obvious. One of the other reasons that I mentioned k-mixup is that I am trying to say that this claim is not novel, in fact, it's easy to be noticed.
>
> To clarify, the claim in our paper is that the distance between pairs of points impacts the likelihood of assigning a wrong label with mixup **and that it impacts calibration**. This is supported in our paper with theoretical and empirical evidence, **both being novel results with a novel approach**.
> Furthermore, the theoretical derivations we show in the paper are more general and with different goal than the ones of k-mixup. k-mixup specifically focus on the regularization effect of mixing samples *matched with optimal transport*, while we study the impact of *random matching* on the label quality. So our results are also novel in that aspect.
>
> > Therefore, if one wants to take this claim as one of the main contributions of their paper, I would prefer to see some more strict justifications. For example, one may mathematically quantify the manifold mismatch behavior and show how the distances of the example pairs may impact this quantity.
>
> We show this in theorem 3.2, **we "mathematically quantify" that the probability of associating a wrong label with mixup is lower bounded by the distance between original class manifolds**. While the result in theorem 3.2 is using the total variation (TV) between the two manifolds, we can further lower bound the TV by the Wasserstein distance between the two manifolds through Theorem 4 in Gibbs and Su (2002) [A], which is more informative of the distance between points within the manifolds.
>
> We also want to clarify that since mixup is applied for multiple iterations (all batch of data seen during training), using random samplings both to determine which data will be mixed and the weights of the interpolations, its effect is averaged over all iterations. That is why we have to study the impact of mixup in terms of probability and distributions.
>
> [A] Gibbs & Su (2002). "On choosing and bounding probability metrics." International statistical review.
>
> > First, the toy example with 3 classes still doesn't fit the claim. Suppose the three classes are three separated line segments on the horizontal axis. If we take: (a) the righest point of class 1 and leftest point of class 2, and (b) the leftest point of class 1 and rightest point of class 3. In this context, case (a) still have much higher chance of having mixed point near original manifolds than case (b).
>
> To be sure that we are on the same page, we considered that in the example, class 1 have been associated to the leftest manifold, class 2 to the middle one, and class 3 to the rightest.
> Then, **that is exactly what we state in the paper** ("the higher the distance between two points, the less likely their convex combination will fall near the original manifolds", l. 238 in the revision). The distance in case (a) is lower than the distance in case (b), and the likelihood of the mixed combination being closer to the original manifolds is higher in case (a) than case (b). For case (b) there is even the possibility to create manifold intrusion, since the mixed sample can fall in class 2, and there is also a portion of the line segment that is closer to class 2 than class 1 or 3. As we stated before, the distance between two points roughly informs about the distance between manifolds. Here, the distance in case (a) being lower than the distance in case (b) tells us that manifolds of class 1 and class 2 are closer than manifolds of class 1 and class 3.
>
> > Second, I personally believe that it's never "limited by-design" to consider some unideal or even worst-case examples of data distribution in algorithm design in general.
>
> There might be a misunderstanding here, we said that the example was "limited by-design" as it had only 2 classes. Conflicts between class manifolds and label associations during mixup, would require either at least 3 class manifolds or convex hulls of manifolds intersecting. We can add $M > 2$ if the reviewer feels that it is necessary.

---

### Official Review · Reviewer_XhfP · 2024-10-31

**Soundness:** 3
**Presentation:** 3
**Contribution:** 3
**Rating:** 6
**Confidence:** 3

**Summary:**

This paper first demonstrates that distance between the data used in mixup can impact the likelihood of manifold mismatch (a phenomenon where mixed samples lie outside of the class manifolds of the original data used for mixing). It then proposes an efficient framework to mitigate the occurrence of manifold mismatch. The key idea is to dynamically change the distributions of mixing coefficients via a similarity kernel that measures the distance between the mixed samples. Empirical results are provided to demonstrate the effectiveness of the proposed method.

**Strengths:**

- Overall the paper is well written and is easy to follow
- An innovative yet efficient method is proposed to improve mixup so that manifold mismatch likelihood is reduced, and better calibration and accuracies can be achieved
- Adequate empirical results on both classification and regression tasks are provided to demonstrate the proposed method

**Weaknesses:**

- While Theorem 1 shows the existence of manifold mismatch, it does not tell us anything about the assumption (which is used in the paper) that the higher the distance between two points, the more likely their convex combination will fall outside of the original manifolds, and thus, the more likely a model would assign a different label than the original labels of the two points. Some theoretical results in establishing the validity of such assumption would further strengthen the paper.
- There are some missing papers that could be worth mentioning in the related work. In particular, mixup related works such as [1,2,3].
- For the experiments, it is not clear how the proposed method would compare with the method proposed by [1,2,3], and also how would it perform for more complex datasets like ImageNet, ImageNet-C, ImageNet-R and ImageNet-P.

---

[1] Dan Hendrycks, Norman Mu, Ekin D. Cubuk, Barret Zoph, Justin Gilmer, and Balaji Lakshminarayanan. AugMix: A simple data processing method to improve robustness and uncertainty. Proceedings of the ICLR, 2020

[2] Soon Hoe Lim, N. Benjamin Erichson, Francisco Utrera, Winnie Xu, and Michael W. Mahoney. Noisy feature mixup. Proceedings of the ICLR, 2022.

[3] Erichson, N. Benjamin, Soon Hoe Lim, Francisco Utrera, Winnie Xu, Ziang Cao, and Michael W. Mahoney. Noisymix: Boosting robustness by combining data augmentations, stability training, and noise injections. Proceedings of the AISTATS 2024.

**Questions:**

- In Eq. (3), there is a -1 inside the exponential. Why not absorb it into $\tau_{max}$? Are there any reasons why it is presented that way?
- In the classification scenario where the data samples (but not the labels) are noisy and corrupted, thus blurring the true distance between the samples used in mixing, how would the proposed method perform?

---

> ### Author Response · Authors · 2024-11-25
>
> We thank the reviewer for their review.
> - **W1**: This is a very insightful remark. We can obtain this result by introducing a new corollary from Theorem 5.1 in Liu et al. [A].
> Let's consider $X, X'$ random variables associated to the inputs, and $\tilde{X} = (1 - \lambda) X + \lambda X'$, for a fixed $\lambda \in [0,1]$. The ground truth conditional distribution of the label $Y$ is expressed as a vector-valued function $f: \mathbb{X} \rightarrow \mathbb{R}^M$, such that $P(Y = m | X = x) = f_m(x)$, for each dimension $m \in \{1, \dots, M\}$. The mixup-induced label can then be assigned based on ground truth $\tilde{Y}^* = arg max_m f_m(\tilde{X})$, or from the combination of conditional distribution $\tilde{Y} = arg max_m [(1 - \lambda) f_m(X) + \lambda f_m(X')]$. With that, we can say that the mixup label is noisy when the two assignments disagree, i.e. when $\tilde{Y}^* \neq \tilde{Y}$.
> Finally, Theorem 5.1 in Liu et al. [A] gives us $P(\tilde{Y}^* \neq \tilde{Y} | \tilde{X}) \geq TV(P(\tilde{Y} | \tilde{X}), P(Y | X))$, where $TV(\cdot, \cdot)$ is the Total Variation.
> We can then derive from Lemma D.1 in Liu et al. [A]:
> $TV(P(\tilde{Y} | \tilde{X}), P(Y | X)) = \frac{1}{2} \sum_{m=1}^{M} | P(Y = m | X) - P(\tilde{Y} = m | \tilde{X}) | = \frac{1}{2} \sum_{m=1}^{M} | f_m(X) - ((1-\lambda) f_m(X) + \lambda f_m(X')) | = \lambda TV(P(Y | X), P(Y' | X'))$
> And symetrically:
> $TV(P(\tilde{Y} | \tilde{X}), P(Y' | X')) = \frac{1}{2} \sum_{m=1}^{M} | P(Y' = m | X') - P(\tilde{Y} = m | \tilde{X}) | = \frac{1}{2} \sum_{m=1}^{M} | f_m(X') - ((1-\lambda) f_m(X) + \lambda f_m(X')) | = (1-\lambda) TV(P(Y | X), P(Y' | X'))$
> Now, we come back to our classification problem with $M$ classes, where samples $(x, y), (x', y') \in (\mathbb{X} \times \{1, \dots, M\})^2$ are drawn from probability measures $\nu_{y}$ respectively supported on the class manifolds $\mathcal{M}\_{y}$. Then, $TV(P(Y | X=x), P(Y' | X'=x'))= TV(\mathcal{M}\_{y}, \mathcal{M}\_{y'})$.
> We can thus formally state that:
> $P(\tilde{Y}^* \neq \tilde{Y} | \tilde{X}) \geq \min (\lambda, (1-\lambda)) TV(\mathcal{M}\_{y}, \mathcal{M}\_{y'})$.
> From all of that we can derive the following insights:
>    - i) The probability of assigning a noisy label is lower bounded by the distance (total variation) between the original manifolds. This gives a theoretical justification for the experiments in Figure 1, since the distance between two points roughly informs about the distance between the two manifolds.
>    - (ii) To avoid noisy labels, we want to reduce the probability that $\lambda = 0.5$ when the original manifolds are far from each other, i.e., when the distance between the two points is high, which is exactly what our SK Mixup is doing.
>
> - **W2 + W3**: We thank the reviewer for the references. We will add the discussions in the related work section.
>   - AugMix [1] improves robustness by combining multiple image augmentations, which makes the approach limited to image data. Furthermore, RegMixup [B] outperforms AugMix, and we show that we can combine our approach with RegMixup to achieve even better robustness results.
>   - NFM and NoisyMix [2,3] both improve robustness by introducing noise before the mixing operations. We think we could combine our method with these approaches, similarly to RegMixup, since the difference seems to not be conflicting. We will investigate that for future work.
>
>   We compared all the methods on ImageNet, ImageNet-R and ImageNet-A in Table 4. We can include more variations of Imagenet for OOD evaluation if the reviewer thinks it is necessary.
>
> - **Q1**: We presented Eq. 3 that way to clearly show that the Gaussian kernel is centered. When the distance between the two points is equal to the average distance, then the interpolation coefficient is sampled from a $\texttt{Beta}(\tau_{max}, \tau_{max})$ distribution.
>
> - **Q2**: In a high level, the distance between two points from two different manifolds roughly informs about the average pairwise distance between the two manifolds. If the noise applied is the same for all samples (like a gaussian noise with fixed mean and variance), and not too high to avoid manifolds intersecting each other, then we expect the method to keep the same benefits.
>
> [A] Liu et al. "Over-training with mixup may hurt generalization." ICLR, 2023
>
> [B] Pinto et al. "Regmixup: Mixup as a regularizer can surprisingly improve accuracy and out distibution robustness." NeurIPS 2022.

---

> > ### Comment · Reviewer_XhfP · 2024-11-26
> > **Thank you for the rebuttal**
> >
> > I thank the author(s) for the responses to my concerns and questions. I am keeping my score.

---

> > > ### Author Response · Authors · 2024-12-01
> > >
> > > Thank you for your feedback, we are glad to know that we have answered all your concerns.

---

### Official Review · Reviewer_ZK3c · 2024-11-04

**Soundness:** 4
**Presentation:** 4
**Contribution:** 3
**Rating:** 8
**Confidence:** 3

**Summary:**

This paper proposes a novel Mixup framework that employs a Similarity Kernel (SK) called SK Mixup to achieve a stronger interpolation between similar points while reducing interpolation otherwise. As a motivation of this study, the authors defined the concept of manifold mismatch, which can negatively impact the calibration of confidence in Mixup. They conducted experimental validation to assess the impact of this phenomenon on the distance between points to mix. Following the presentation of SK Mixup, the effectiveness of the proposed approach in alleviating the manifold mismatch was demonstrated through extensive experiments in classification and regression. Nevertheless, the extension to more complex tasks and theoretical analysis on the regularization effect of SK Mixup still need to be undertaken -- This is why the authors did not receive the highest score in *Contribution*.

**Strengths:**

* The proposed approach was well motivated by targeting the appropriate research problem of calibration-driven Mixup methods in classification and regression: Manifold mismatch. Furthermore, this problem was theoretically defined and proved with the supplement materials.

* The proposed method was in a clear and well-organized manner, and the proof was provided in a manner that supported its conclusions.

* The research problem was validated through experimental results; a comparison was made with the baseline.

* Comprehensive experiments in classification and regression, with a particular focus on performance and computational cost, demonstrated the efficacy of the proposed approach.

**Weaknesses:**

There is no weakness in the paper that would justify its rejection.

**Questions:**

**Question 1.** Why did not SK RegMixup be applied to ViT in Table 4? Is there any potential challenges or limitations of applying SK RegMixup to ViT architectures?

**Question 2.** [line 976-978] In the proof of the first part in Theorem 3.1., whatever the value of $\lambda\_{1}$ is, $\lambda$ can be 0. When $\lambda=0$, then $\tilde{\mathbf{x}}(\lambda)=\mathbf{x}\_{l} \in \mathcal{M}\_{j}$, not $\mathcal{M}\_{i}$, and so does symmetrical case. In my opinion, it would be $\forall \lambda \geq \lambda\_{1},~\tilde{\mathbf{x}}(\lambda)=\lambda \mathbf{x}\_{k}+(1-\lambda)\mathbf{x}\_{l} \in \mathcal{M}\_{j}$. The authors are kindly requested to examine and fix them if there is an error.

---

**Things to improve the paper that did not impact the score:**

* Figure 4 caption: there is no result about *Circles* toy datasets. The mention of it from the caption would be removed if it's not intended to be included

* [line 442] The ECE and AECE of MIT-A in ImageNet-R would be in bold. The authors have to double-check their result highlighting in Table 4 to ensure consistency and accuracy.

*  In Table 4, is there any reason why ECE and AECE are exactly same in OOD settings? The authors are kindly requested to explain why ECE and AECE are identical in OOD settings or, if this is unexpected, verify whether there might be an error in the reporting or calculation of these metrics.

* The experimental results show that the SK RegMixup is an effective method. However, Table 6 does not provide the efficiency of the SK RegMixup in terms of computational cost. It is recommended to include computational cost metrics for SK RegMixup in Table 6, similar to what they've provided for other methods.

* [line 879] Typo: cccross

---

> ### Author Response · Authors · 2024-11-25
>
> We thank the reviewer for their review and appreciate the kind feedback. We answer below to your questions and remarks.
>
> * **Q1**: There are no inherent problems to apply calibration-driven mixup methods to ViTs, the only limitation is the computation time to have a full benchmark. Here is a more complete table of results for the ViT-S/16 architecture, on ImageNet (IN), ImageNet-R (IN-R) and ImageNet-A (IN-A), that we will include in the revision:
>
>   Methods | IN Acc | IN ECE | IN AECE | IN-R Acc | IN-R ECE | IN-R AECE | IN-A Acc | IN-A ECE | IN-A AECE
>    --- | --- | --- | ---- | --- | --- | ---- | --- | --- | ----
>   ERM | 69.34 | 9.03 | 9.03 | 15.46 | 35.93 | 35.93 | 1.83 | 45.03 | 45.03
>   Mixup | 72.0 | 5.81 | 5.8 | 18.21 | 29.29 | 29.29 | 2.47 | 41.07 | 41.07
>   Manifold Mixup | 72.04 | 6.99 | 6.95 | 18.93 | 30.27 | 30.27 | 2.15 | 43.47 | 43.47
>   RegMixup | _74.44_ | 7.36 | 7.34 | _21.01_ | 32.64 | 32.64 | _3.8_ | 44.4 | 44.4
>   RankMixup | 69.99 | 9.3 | 9.3 | 15.77 | 37.37 | 37.37 | 1.77 | 46.82 | 46.82
>   MIT-A | 72.81 | 5.69 | 5.69 | 17.6 | 31.07 | 31.07 | 2.81 | 41.8 | 41.8
>   SK-Mixup (Ours) | 71.83 | _3.89_ | _3.9_ | 18.37 | _28.56_ | _28.56_ | 2.28 | _39.33_ | _39.33_
>   SK RegMixup (Ours) | **75.11** | **2.0** | **1.98** | **22.06** | **26.23** | **26.23** | **4.44** | **38.25** | **38.25**
>
> * **Q2**: This is partially correct. We wanted to separate the definitions of $\lambda_1$ and $\lambda_2$ for clarity, but we overlooked that they need to be defined jointly. We will change this part in the proof by defining both jointly:
> "On the line segment $[x_k, x_l]$, since $z \in \mathcal{M}_i$ and manifolds are disjoint , $\exists (\lambda_1, \lambda_2) \in [0,1]^2, \lambda_1 > \lambda_2$ such that $\forall \lambda \in ]\lambda_2, \lambda_1[, \tilde{x}(\lambda) = \lambda x_k + (1-\lambda) x_l \notin \mathcal{M}_j$. Then, $\lambda_0 \in ]\lambda_2,\lambda_1[$."
>
> * > In Table 4, is there any reason why ECE and AECE are exactly same in OOD settings?
>
>    The only difference between the computations of ECE and AECE is the separation into the bins. AECE dynamically defines the range of the bins to have evenly distributed number of samples within each bin, while ECE fixes the range of each bin. Thus, if the bins are already close to evenly distributed, ECE and AECE will give similar results. We think that it might be more likely to happen in OOD settings. We refer the reviewer to Table 10 in the appendix, reporting AECE in ID settings, in which the values are different than ECE.
>
> * > Table 6 does not provide the efficiency of the SK RegMixup in terms of computational cost.
>
>    Here is a more complete comparison of computational costs for baselines considered for Image Classification. We will include these in the table in the revision of the paper.
>
> Method | Batch in memory | C10 - Best epoch | C10 - Time / epoch (s) | C10 - Total Time (s) | C100 - Best epoch | C100 - Time / epoch (s) | C100 - Total Time (s) | Tiny-IN - Best epoch | Tiny-IN - Time / epoch (s) | Tiny-IN - Total Time (s)
> --- | --- | --- | --- | --- | --- | --- | --- | --- | --- | ---
> Mixup | 1 | 186 | 17 | 3162 | 181 | 17 | 3077 | 89 | 125 | 11125
> RankMixup | 4 | 147 | 78 | 11485 | 177 | 78 | 13806 | 80 | 540 | 43380
> MIT-A | 2 | 189 | 46 | 8694 | 167 | 46 | 7659 | 82 | 305 | 24908
> RegMixup | 2 | 173 | 32 | 5536 | 177 | 32 | 5664 | 94 | 240 | 22560
> SK Mixup | 1 | 183 | 28 | 5138 | 160 | 28 | 4480 | 69 | 189 | 13104
> SK RegMixup | 2 | 175 | 39 | 6825 | 179 | 39 | 6981 | 75 | 316 | 23700

---

> > ### Comment · Reviewer_ZK3c · 2024-11-28
> >
> > **Acknowledgement to the Authors**
> >
> > I am grateful for the supplementary efforts and the clarification. Also, thanks for considering and answering the minor things.
> >
> >
> > **Closing remarks**
> >
> > My research interests in the research problem (defined as manifold mismatch in this paper) and the goal (effective control of the mixup rate to improve the quality of mixed samples) are aligned with the ones of the authors. Therefore, this probably resulted in a higher score than others.
> >
> > To adopt an objective perspective, I perused all other reviews and comments. I concede that I may have made a bit of an impetuous judgment by taking certain details for granted, such as the connection, referred to in reviewer XhfP, between manifold mismatch and the authors' assumption.
> >
> > Nevertheless, in light of the revisions, where the authors responded to the reviews comprehensively by incorporating additional experimental results and a theorem, and the strengths previously indicated in my review, I still believe this paper deserves a positive rating. Consequently, I will maintain my original rating.
> >
> > However, it is essential to recognize the potential for bias in my review, which should be taken into account in the final decision.

---

> > > ### Author Response · Authors · 2024-12-01
> > >
> > > Thank you very much for your detailed and positive feedback. We are glad that you appreciated our efforts to address all reviewers' comments with the additions in the revision.

---

### Comment · Reviewer_Fzdf · 2024-11-22

My apologies, I mistook the ending date of the discussion period. I have undone the changes I had made.

---

### Author Response · Authors · 2024-11-25
**Revision of the paper and summary of changes**

We thank all the reviewers for their reviews and their insightful comments about our work. We uploaded a revision of the manuscript to include your suggestions. All modifications in the text are colored in blue. Here is a summary of the changes:
- We improved the presentation and clarified some sentences.
- We combined old Figure 1 and Figure 2 into a single Figure using subfigures.
- We added a novel theoretical result (Theorem 3.2) supporting our experiments in Figure 1a, that we derived as a corollary of Theorem 5.1 from Liu et al. [A]. This result shows that the likelihood of assigning a wrong label with Mixup is lower-bounded by the distance (total variation) between the original manifolds. The proof is in Appendix F.2.
- We added missing baselines for ViT architectures on Imagenet in Table 4.
- We added in Appendix L a comparison with CutMix [B] and with discussions around combining our method with CutMix.
- We added time comparison with all baselines in Table 6 for Image classification. The full comparison on regression tasks and with non-linear mixup (CutMix) was moved to Appendix M

[A] Liu et al. "Over-training with mixup may hurt generalization." ICLR, 2023

[B] Yun et al. Cutmix: Regularization strategy to train strong classifiers with localizable features. In ICCV 2019.

---

### Meta-Review · Area_Chair_sojo · 2024-12-23

**Metareview:**

This paper introduces a new Mixup method that is able of improve both accuracy and calibration. Specifically, the authors propose a Similarity Kernel to parameterize the distribution of interpolation coefficients, making Mixup more data-centric. The idea is both novel and significant. Extensive experiments demonstrate that this approach improves accuracy and calibration across a variety of tasks, including classification and regression.

The reviewers are positive, noting that the method is well motivated, theoretically justified, and supported by strong experimental validation. Major concerns were addressed during the rebuttal phase, and the paper is clear and enjoyable to read. One possible extension is to consider a non-linear or latent space similarity kernel, for instance, employing a pre-trained autoencoder to compute similarity in latent space.

I recommend to accept this submission.

**Additional Comments On Reviewer Discussion:**

The authors have carefully addressed all major concerns during the rebuttal phase and revised the manuscript accordingly. Also, the paper has greatly improved compared to a previous version that I have seen.

---

### Decision · Program_Chairs · 2025-01-22

Accept (Poster)